RESEARCH COMMUNICATION

# Sox9+ messenger cells orchestrate large-scale skeletal regeneration in the mammalian rib

Stephanie T Kuwahara[1], Maxwell A Serowoky[1], Venus Vakhshori[2], Nikita Tripuraneni[1], Neel V Hegde[1], Jay R Lieberman[2], J Gage Crump[1], Francesca V Mariani[1]*

[1]Department of Stem Cell Biology and Regenerative Medicine, University of Southern California, Keck School of Medicine, Los Angeles, United States; [2]Department of Orthopaedic Surgery, University of Southern California, Keck School of Medicine, Los Angeles, United States

**Abstract** Most bones in mammals display a limited capacity for natural large-scale repair. The ribs are a notable exception, yet the source of their remarkable regenerative ability remains unknown. Here, we identify a *Sox9*-expressing periosteal subpopulation that orchestrates large-scale regeneration of murine rib bones. Deletion of the obligate Hedgehog co-receptor, *Smoothened*, in *Sox9*-expressing cells prior to injury results in a near-complete loss of callus formation and rib bone regeneration. In contrast to its role in development, Hedgehog signaling is dispensable for the proliferative expansion of callus cells in response to injury. Instead, Sox9-positive lineage cells require Hh signaling to stimulate neighboring cells to differentiate via an unknown signal into a skeletal cell type with dual chondrocyte/osteoblast properties. This type of callus cell may be critical for bridging large bone injuries. Thus despite contributing to only a subset of callus cells, *Sox9*-positive progenitors play a major role in orchestrating large-scale bone regeneration.

**Editorial note:** This article has been through an editorial process in which the authors decide how to respond to the issues raised during peer review. The Reviewing Editor's assessment is that all the issues have been addressed (see decision letter).

DOI: https://doi.org/10.7554/eLife.40715.001

*For correspondence:
fmariani@usc.edu

**Competing interests:** The authors declare that no competing interests exist.

## Introduction

Whereas amphibians regenerate large portions of their skeletons after injury or amputation, natural large-scale skeletal repair in mammals is more limited. A notable exception is the rib. Craniofacial surgeons often extract large segments of rib cartilage and bone for autologous repair of skeletal defects in other parts of the body, and in post-operative visits have noted extensive regeneration at the donor rib site (*Kawanabe and Nagata, 2006*; *Munro and Guyuron, 1981*; *Srour et al., 2015*). To better understand the unique regenerative potential of the rib, we have recently developed analogous large-scale rib cartilage and bone regeneration models in the mouse (*Srour et al., 2015*; *Tripuraneni et al., 2015*).

In most mammalian bones, healing involves the formation of a callus using a mixture of direct ossification and endochondral ossification via a cartilage callus intermediate (*Gerstenfeld et al., 2003*; *Hall, 2014*; *Marsell and Einhorn, 2011*). The identity and regulation of the adult skeletal progenitors that build the callus remain incompletely understood. While markers for a number of different skeletal stem cells have been reported (*Balani et al., 2017*; *Bianco and Robey, 2015*; *He et al., 2017*; *Matthews et al., 2014*; *Park et al., 2012*; *Ransom et al., 2016*; *Shi et al., 2017*), their

**eLife digest** Fractures to major bones often heal slowly or incompletely, especially in older people, and large bone injuries do not repair naturally. By comparison, rib bones show an unusual capacity to regrow and repair themselves even when a large portion is damaged. Previous research suggests that the connective tissue around the ribs helps to support and co-ordinate bone healing. Yet it is currently not clear why ribs have a greater capacity to repair these large injuries compared to other bones.

Kuwahara et al. have now examined bone repair by studying how ribs heal in mice. The experiments show that around 6% of the connective tissue cells are critical for large-scale repair. Kuwahara et al. named these cells messenger cells. These cells detect the presence of a signal molecule called Hedgehog (Hh), however, if these cells lose the ability to respond to the Hh molecules, the ribs do not heal properly. Further examination revealed that these messenger cells co-ordinate repair by encouraging other cells to build a special kind of bone-healing tissue with hybrid properties of both cartilage and bone.

Further research could now examine how messenger cells co-ordinate healing and if their properties could be adapted to help repair other bones. Ultimately, understanding how messenger cells work may even provide insights into new ways to repair and regenerate other tissues and organs too.

DOI: https://doi.org/10.7554/eLife.40715.002

relative roles during bone repair are less clear, particularly in cases of large-scale bone regeneration such as in the rib. During skeletal repair, studies have shown that cells from the periosteum, a heterogeneous connective tissue sheath covering the bone, are major contributors to the callus (*Colnot, 2009*; *Duchamp de Lageneste et al., 2018*; *Murao et al., 2013*). Accordingly, we have found that the periosteum is essential for regeneration of the rib bone (*Tripuraneni et al., 2015*).

During normal bone homeostasis, periosteal stem cells generate bone-producing osteoblasts but not cartilage-producing chondrocytes (*Roberts et al., 2015*). How injury stimulates periosteal stem cells to generate chondrocytes is unclear. Although some fractures can heal in the absence of a cartilage callus, for example when the fracture is rigidly stabilized (*Thompson et al., 2002*), the formation of a large cartilage callus appears to be required in large-scale bone regeneration. Unfortunately little is known about the specific periosteal progenitor population that drives the formation of the cartilage callus nor the signaling pathways required. Recent studies have used Cre-based lineage tracing experiments to show that cells marked by expression of *Gremlin1* (*Worthley et al., 2015*), *Axin2* (*Ransom et al., 2016*), *Gli1* (*Shi et al., 2017*), *Act2a* (*Matthews et al., 2016*), *Periostin* (*Duchamp de Lageneste et al., 2018*) and *Sox9* (*Balani et al., 2017*; *He et al., 2017*) can be found in the periosteum and contribute to the fracture callus during repair. Other than participation, the specific role of any of these progenitor population remains unclear. In this study, we therefore focus on the role of one subpopulation within the periosteum and its specific role in driving callus formation and bone regeneration. As *Sox9* has a well-known function in promoting chondrogenesis during embryonic development (*Akiyama et al., 2002*; *Lefebvre et al., 1997*), we postulated that *Sox9*-expressing cells in the adult periosteum may possess a potent ability to form or promote cartilage in response to bone injury. In addition, a requirement for *Sox9*-expressing cells in repair or homeostasis has not yet been established.

The signaling pathways required for generating a cartilage callus from the periosteum in response to injury are also not well-characterized. The formation of the cartilage callus in fractures has been thought to involve a recapitulation of endochondral bone development (*Bahney et al., 2014*; *Maes et al., 2010*; *Marsell and Einhorn, 2011*; *Yang et al., 2014*; *Zhou et al., 2014*). One of the most well-known signaling pathways during endochondral bone development is the Hh pathway, activation of which promotes chondrogenic proliferation and osteocyte maturation (*Long et al., 2001*; *Shi et al., 2017*; *St-Jacques et al., 1999*). Studies have investigated Hh signaling during fracture repair, but due to conflicting results it remains unclear if Hh signaling has the same role in bone repair as it does during bone development. Genetic ablation of the obligate Hh co-receptor *Smo* in mice, using two different ubiquitously inducible Cre lines, resulted in reduced bone formation during

fracture repair, yet was not reported to disrupt initial cartilage callus formation (*Baht et al., 2014*; *Wang et al., 2010*). Forced activation of Hh signaling throughout the mouse during fracture repair, using an inducible constitutively active *Smo* allele, resulted in increased bone formation (*Baht et al., 2014*), similar to what was seen upon engraftment of cells overexpressing Hh or treatment with an Hh agonist (*Edwards et al., 2005*; *Huang et al., 2014*; *Zou et al., 2014*). However, on which cell types Hh acts upon, and whether it regulates the decision to build the cartilage callus and/or other aspects of bone repair in mammals, has remained unknown.

In this study we examine the role of the *Sox9*-expressing periosteal subpopulation (referred to as Sox9+ cells hereafter) during large-scale rib repair. This subpopulation appears to be a key player in orchestrating the formation of a large repair callus in the rib that consists of an unusual hybrid osteochondral cell type with properties of both chondrocytes and osteoblasts. Loss of *Smo* in Sox9+ periosteal cells prior to injury results in a near-complete failure of cartilage callus formation and bone regeneration. This Sox9+ subpopulation must be able to respond to Hh signaling in order to initiate this process, indicating that Hh signaling's role in bone repair is distinct from its role in bone development. Additionally, since Sox9+ periosteal cells contribute to only a minority of callus cells, we suggest that Sox9+ periosteal cells act as 'messenger' cells and orchestrate repair by inducing the differentiation of neighboring callus cells through non-autonomous signals. Overall our results indicate that bone regeneration does not fully recapitulate bone development, and that the periosteum consists of subpopulations that may have different roles/responses during repair.

## Results

### The murine rib bone regenerates through a hybrid skeletal cell type

Like appendicular long bones, the bony portion of the rib develops via an endochondral process including growth plates at either end and a central hollow bone marrow cavity. Both human and murine rib bones display remarkable regenerative potential (*Srour et al., 2015*; *Tripuraneni et al., 2015*), however the cellular basis for such large-scale repair remains unknown. To better understand the cellular sequence of events during regeneration, we analyzed 3 mm rib bone defects at sequential time points up to 10 weeks post-resection (wpr) (*Figure 1A*). Histology at 5 days post-resection (dpr) revealed cells with a mesenchymal-like morphology filling the entire resected region (*Figure 1B*). We then observed formation of a substantial alcian-blue positive callus spanning most of the defect by 1 wpr (*Figure 1A*), with many of these cells displaying a cartilage-like morphology at 10 dpr (*Figure 1C*). Histology revealed increasing bone formation by 10 and 14 dpr (*Figure 1C, D*), with extensive alizarin-positive mineralization across the defect at 4 wpr and full remodeling to the pre-injury organization by 10 wpr (*Figure 1A*).

Next, we used double fluorescent RNA in situ hybridization (RNA-ISH) to characterize the molecular identity of cells during the regeneration process. At 5 dpr, mesenchymal cells within the lesion co-expressed *Sox9*, a master regulator of the chondrocyte lineage (*Akiyama et al., 2002*; *Lefebvre et al., 1997*), and *Runx2*, a master regulator of the osteoblast lineage (*Lian and Stein, 2003*) (*Figure 1B*). While co-expression of chondrocyte and osteoblast markers has been observed during early bone development, normally as differentiation proceeds, a cell will express only chondrocyte or only osteoblast markers. Surprisingly, we observed co-expression of genes associated with chondrocyte and osteoblast differentiation within single cells throughout the repair process. For example, we see co-expression of *Sox9* with the major osteoblast collagen gene *Col1a1* (*Figure 1—figure supplement 1A*). As the callus matures bone and cartilage markers continue to be co-expressed at high levels. The major chondrocyte collagen gene *Col2a1* was observed to be co-expressed with the late osteoblast marker *Bglap* (also known as *Osteocalcin*) (*Figure 1C* and *Figure 1—figure supplement 1C*), and with *Col1a1* at both 10 and 14 dpr (*Figure 1D* and *Figure 1—figure supplement 1B*). We observed co-expression of chondrocyte and osteoblast markers in cells of both cartilage and osteoblast morphology, including cells on the surface of newly formed trabecular bone (*Figure 1C,D*, *Figure 1—figure supplement 1C*). Similar co-expression results were obtained using mice double transgenic for a reporter of hypertrophic chondrocytes (*Col10a1-mCherry*) and osteoblasts (*Col1(2.3)-GFP*) as well as using double immunofluorescence for COL1 and COL2 protein (*Figure 1E*, *Figure 1—figure supplement 1D*). In contrast, we did not observe co-expression of *Col2a1* and *Col1a1* in the rib growth plate under the same assay conditions

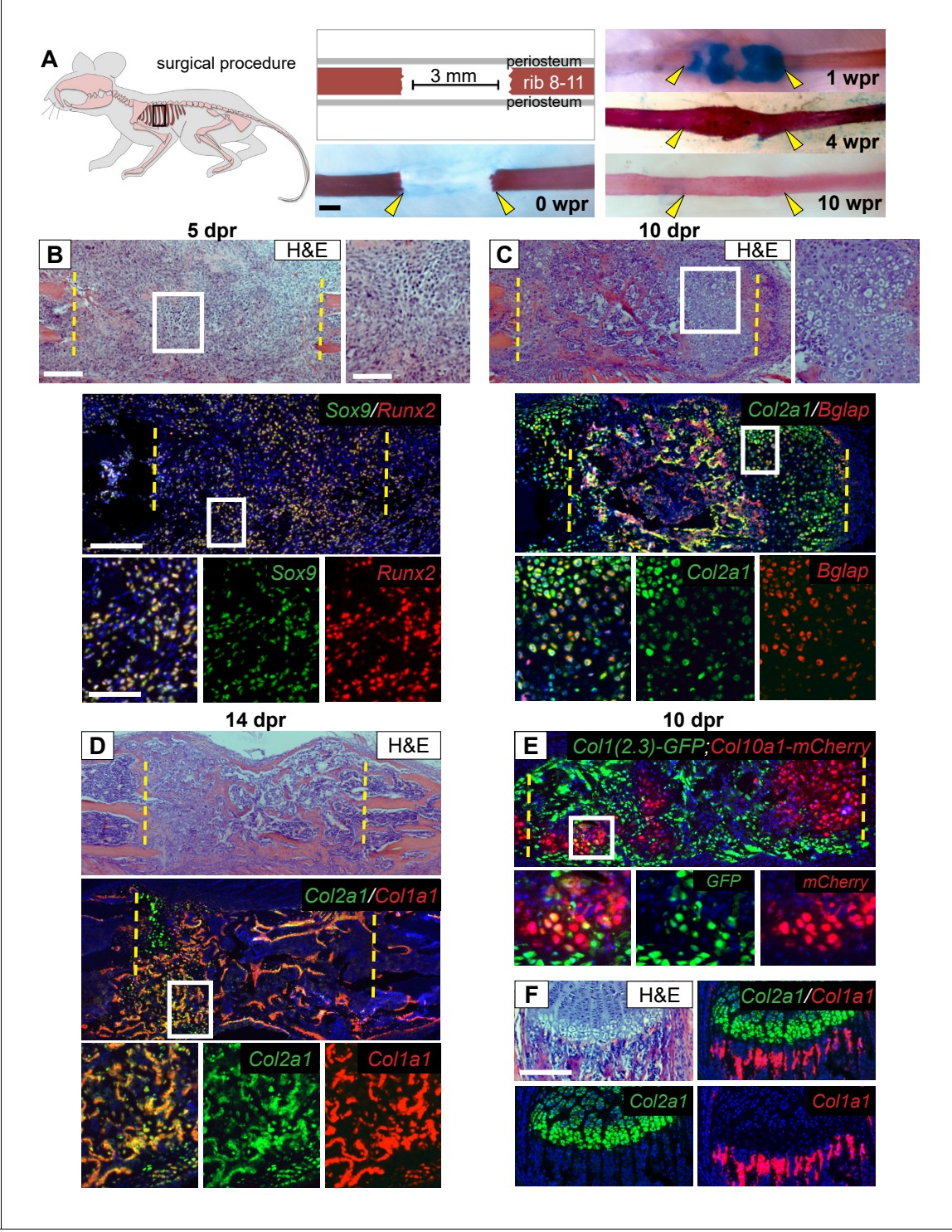

**Figure 1.** Regeneration involves skeletal cells with hybrid osteochondral properties. (A) Schematic of the murine rib resection model. A 3 mm bone segment is resected from one rib (8-11), while the periosteum is carefully released and left in the mouse. Alizarin red and alcian blue whole mount staining indicates that repair occurs through a cartilage intermediate. The images show the outcome immediately after the resection at 0 wpr (weeks post resection, n = 2). At 1 wpr alcian blue positive material is evident between the cut ends (n = 3), by 4 wpr (weeks post resection) the lesion is fully-spanned by a mineralized callus (n = 2), while by 10 wpr remodeling has occurred (n = 2). (B–D) Histological sections stained with hematoxylin and eosin (H and E) (n > 5 for each time point) and near-adjacent double fluorescent RNA in situ hybridization (RNA-ISH) assays confirm the presence of a cartilage intermediate and show expression patterns in the repair callus. (B) At 5 dpr (days post resection) mesenchymal-like progenitor cells have moved into the resected region and are positive for the expression of both *Sox9* (green) and *Runx2* (red). The enlarged boxes with the separated color

*Figure 1 continued on next page*

Figure 1 continued

channels show co-expression in many of the cells within the resected region (overlap is yellow) (n = 3). (C) By 10 dpr bone and cartilage formation spans the resected region. Many cells that mediate repair express both the chondrocyte-associated gene *Col2a1* (green), as well as the osteoblast-associated gene *Bglap* (red). The enlarged boxes show cells that have chondrocyte morphology expressing both *Col2a1* and *Bglap* (n = 3). (D) At 14 dpr trabecular bone spans almost the entire resected region with only a small amount of cartilage at the cut ends; cells expressing both *Col2a1* (green) and *Col1a1* (red) are widespread. The enlarged boxes show the surface of newly formed trabecular bone where cells can be found that co-express *Col1a1* and *Col2a1* (n = 4). (E) At 10 dpr, animals double transgenic for *Col1(2.3)-GFP;ColX-mCherry* have mCherry (red) positive cells that are also expressing the osteoblast-specific reporter for *Col1* (green). (F) Expression of *Col2a1* (green) and *Col1a1* (red) of the rib growth plate from an uninjured animal does not show a high degree of overlap (n = 5). *Col2a1* is highly expressed in chondrocytes of the growth plate but not in osteoblasts forming new bone, while *Col1a1* is highly expressed in the osteoblasts/cytes below the growth plate but not in cartilage cells. Scale bar for A = 500 microns, B-F = 200 microns, enlarged box H and E = 100 microns, enlarged box = 50 microns.

DOI: https://doi.org/10.7554/eLife.40715.003

The following figure supplements are available for figure 1:

**Figure supplement 1.** Analysis of hybrid skeletal cells.

DOI: https://doi.org/10.7554/eLife.40715.004

**Figure supplement 2.** Colormetric RNA-ISH.

DOI: https://doi.org/10.7554/eLife.40715.005

(*Figure 1F*). Single colorimetric RNA-ISH assays of near-adjacent sections at 7 dpr confirmed co-expression of *Sox9*, *Runx2*, *Col2a1*, *Col1a1*, *Bglap*, and the hypertrophic cartilage collagen gene *Col10a1* in cells of cartilage morphology within the callus (*Figure 1—figure supplement 2*). Together, these findings indicate that, in marked contrast to the growth plate, cells within the rib repair callus co-express cartilage and bone markers while displaying either a chondrocyte or osteoblast morphology. We therefore refer to these cells as 'hybrid' osteochondral skeletal cells.

## Lineage tracing of *Sox9*-expressing cells to the regenerating rib callus

We next examined the source of the skeletal cells that mediate repair. These cells are likely derived from the periosteum, since removal of the periosteum along with the bone, results in a failure of cartilage callus formation (*Figure 2—figure supplement 1*) and subsequent bone repair (*Tripuraneni et al., 2015*). *Sox9* is a master regulatory gene of chondrogenesis, and a previous study indicated that Sox9+ cells in the femur periosteum can contribute to callus formation after fracture (*He et al., 2017*). We therefore examined whether Sox9-expressing periosteal cells contribute to the repair callus during rib regeneration. To do so, we administered three consecutive daily doses of tamoxifen to *Sox9-CreERT2; ROSA26-loxP-stop-loxP-tdTomato* mice, followed by a 4-day chase to allow clearance of residual tamoxifen (*Figure 2A*, "Pre" regimen). In the absence of injury, we observed that *Sox9*-expressing cells were predominantly found in the periosteum and but only constituted 6 ± 0.3% of the population, with almost no labeled cells seen in the marrow compartment (*Figure 2B*). After rib resection, we found that 22 ± 1.3% of callus cells were tdTomato+ by 10 dpr with many of these cells having a typical chondrocyte morphology (*Figure 2C*). At 14 dpr (*Figure 2C*), we observed tdTomato expression in both chondrocyte-like cells, as well as osteoblast-like cells lining new trabecular bone (as defined by near-adjacent H&E sections), with contribution to osteoblast-like cells also evident by 21 dpr (*Figure 2—figure supplement 2A*). We observed fewer tdTomato+ cells at 14 dpr suggesting that as the callus matures, Sox9+ lineage cells are remodeled out and replaced by non-Sox9+ lineage cells. Pre-existing -expressing cells, thus contribute to only a minority of the cells that form the initial repair callus, including only a subset of the callus cells with hybrid osteochondral characteristics. Most of the cells in the rib callus are therefore derived from a lineage that did not express *Sox9* prior to injury. In addition, similar to reports by He *et al.* (*He et al., 2017*) we found that also after femur fracture, our "Pre" tamoxifen regimen revealed contribution of pre-existing *Sox9*+ cells to the callus (*Figure 2—figure supplement 3A*). We further found that some of the cells within the femur fracture callus also co-expressed *Col1a1* and *Col2a1* strongly, suggesting a similar hybrid identity to that observed in the rib callus (*Figure 2—figure supplement 3B*).

Since only a minority of the callus cells are derived from the *Sox9*-expressing periosteal subpopulation, as opposed to the majority of the callus cells that expresses *Sox9* mRNA at 5 dpr (*Figure 1B*), we instead applied tamoxifen at the time of injury plus 2 days following ("Post" regimen,

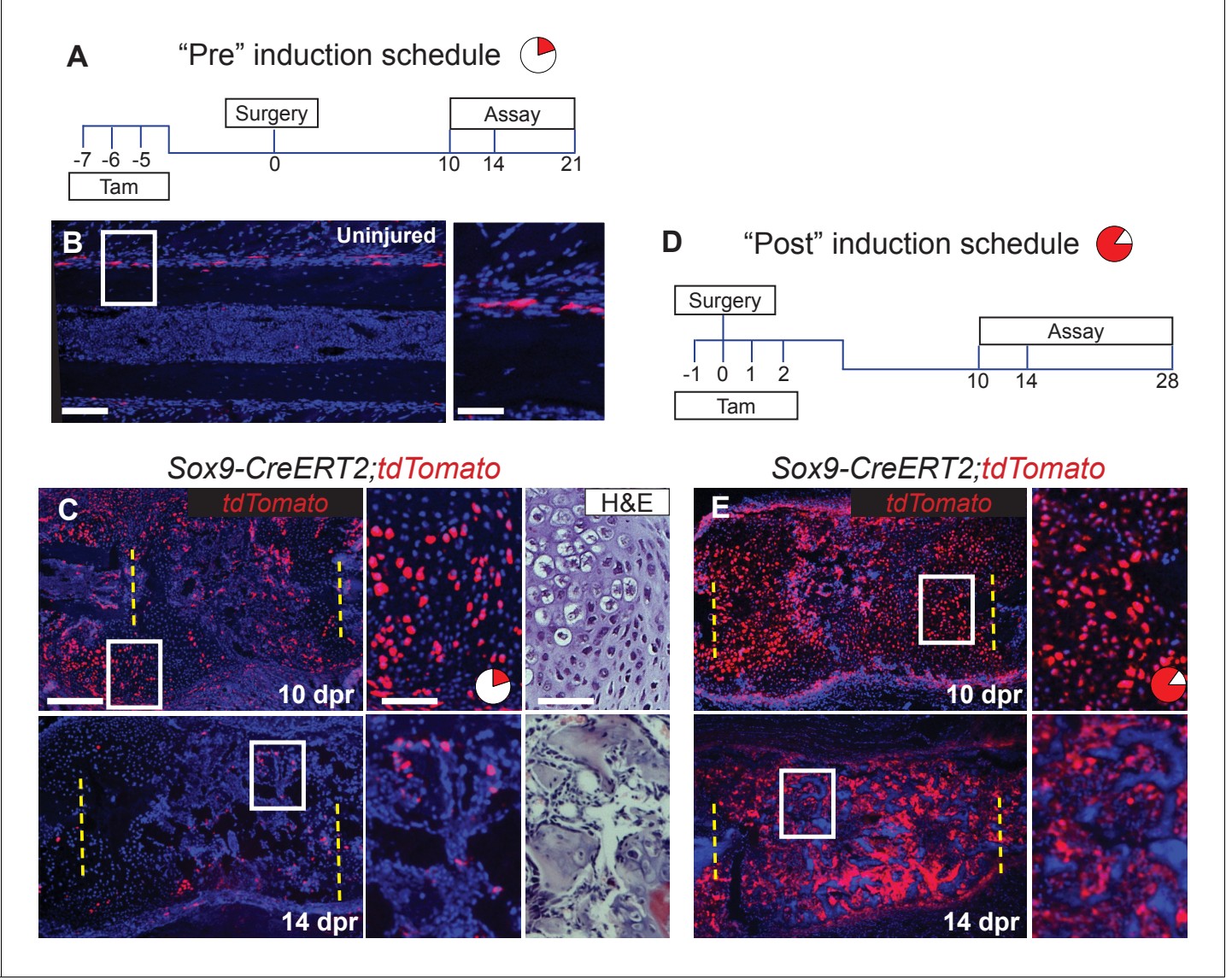

**Figure 2.** *Sox9-CreERT2* marks cells that participate in repair. (**A**) To target a sub-population of callus cells that arise from the periosteum, *Sox9-CreERT2;tdTomato* mice were injected for 3 consecutive days starting 7 days before analysis or before surgery. This tamoxifen induction schedule is referred to as 'Pre'. The pie chart is a visual representation of number of cells affected in the callus at 10 dpr (red) and will be used throughout the figures as a reminder. (**B**) In uninjured *Sox9-CreERT2;tdTomato* mice, Sox9+ cells can be observed within the periosteum of the diaphysis by immunofluorescence (IF) for the tdTomato protein (red) (n = 3). They make up 6 ± 0.3% of the periosteal cells. (**C**) To determine if these cells participate in repair, rib resections were performed. At 10 dpr, lineage tracing of the Sox9+ cells show that the tdTomato+ expressing cells, 22 ± 1.3%, contribute to both cartilage and bone (n = 3) (IF for tdTomato). The enlarged images show chondrocytes, some of which are positive for tdTomato along with a near-adjacent section showing the histology of the area. At 14 dpr, tdTomato-expressing cells can be seen contributing to the trabecular bone. The enlarged image shows cells lining the trabeculae that are positive for tdTomato. (**D**) To activate Cre in a larger percentage of cells that build the repair callus, *Sox9-CreERT2;tdTomato* mice were injected with tamoxifen for 4 consecutive days, starting the day before surgery took place, the day of surgery, and for 2 more days. This tamoxifen induction schedule is referred to as 'Post'. Another pie chart is used to visualize that a much greater portion of the callus is affected using this tamoxifen induction schedule at 10 dpr (85.5 ± 2.8%). (**E**) At 10 dpr, tdTomato+ cells are present in developing cartilage and bone, as well as in the periosteum surrounding the callus (native tdTomato fluoresence from a cryosection). The enlarged panel shows cells with chondrocyte morphology that are tdTomato+. At 14 dpr, tdTomato+ cells can be seen building new trabecular bone. In comparison to the Pre induced mice, these mice have significantly more tdTomato+ cells within the repair callus (n > 2 for all time points). Scale bar B = 100 microns, enlarged box = 50 microns; C, E = 200 microns.

DOI: https://doi.org/10.7554/eLife.40715.006

The following figure supplements are available for figure 2:

**Figure supplement 1.** Periosteal removal.

*Figure 2 continued on next page*

*Figure 2 continued*

DOI: https://doi.org/10.7554/eLife.40715.007

**Figure supplement 2.** Lineage tracing at later time-points.

DOI: https://doi.org/10.7554/eLife.40715.008

**Figure supplement 3.** Femur fracture.

DOI: https://doi.org/10.7554/eLife.40715.009

*Figure 2D*). This regimen will capture both the subpopulation that expresses *Sox9* prior to injury as well as any cells that turn on *Sox9* in response to injury. Consistent with detection of endogenous *Sox9* mRNA expression in early callus cells (e.g. *Figure 1B*), we found that this later tamoxifen regimen labeled the majority of callus cells by 10 dpr (85.5 ± 2.8%) and most of the osteoblast-like cells lining newly forming trabecular bone at 14 (*Figure 2E*) and 28 dpr. Further, typical of osteocytes, some of the tdTomato+ cells at 28 dpr were found embedded in bone (*Figure 2—figure supplement 2B*). Thus, depending on when tamoxifen is administered (Pre vs. Post), either a minority or majority of the callus cells are labelled.

## Requirement of Hh signaling in *Sox9*-expressing cells for callus formation

Since we hypothesized that the Hh pathway may be important for large-scale repair, we first determined the expression of Hh ligands and *Ptch1* after rib resection. As expected, based on the expression of *Ihh* in the growth plate, callus cells with cartilage morphology expressed *Ihh* and *Ptch1*. Interestingly we also detected the related ligand, *Shh*, in these cells (*Figure 3A and C*). At earlier stages *Ihh* expression was undetectable until 7 dpr and then only at very low levels in developing chondrocytes. However, *Shh* expression was readily detectable at 5 dpr and 7dpr similar to reports by others (*Matsumoto et al., 2016*) although at lower levels than found in cells with cartilage morphology (*Figure 3A and C*). *Ptch1*, a read-out of the Hh pathway, was also detected in tdTomato + periosteal cells a 3 dpr, supporting the idea that these cells are responsive to a Hh signal prior to entering the callus (*Figure 3—figure supplement 1A*).

We then tested the requirement for Hh signaling in large-scale bone regeneration by deleting *Smo*, the required Hh co-receptor, using the *Sox9-CreERT2* transgene and employing either the Pre or Post tamoxifen treatment regimens. Using *Smo* RNA-ISH, we observed that the Post tamoxifen treatment resulted in deletion of *Smo* broadly (*Figure 3B*) as predicted given that Cre is active in most cells (*Figure 2D*). Pre tamoxifen treatment resulted in deletion of *Smo* in the Sox9-positive lineage subpopulation which were marked by including the tdTomato reporter in the cross (*Figure 3B*). This was expected, since deletion of *Smo* occurred prior to surgery and thus only a minority of the cells in the repair callus are expected to be null for *Smo* (*Figure 2C*).

We then examined the callus at 7 dpr while there is a mix of both immature and differentiating cells. In control Pre tamoxifen treatment animals, tdTomato+ cells could be found scattered across the developing callus and were not preferentially found in clusters of differentiating chondrocytes suggesting that Sox9+ cells do not differentiate ahead of other cells in the callus (*Figure 3—figure supplement 1B*). In addition, in regions of the control callus where cells appeared immature and in Pre tamoxifen treatment calluses, there was no strong correlation between the tdTomato trace and the expression of *Shh*, or read-outs of Hh signaling such as *Ptch1* or *Gli2* (*Pak and Segal, 2016*) suggesting that in contrast to our earlier observations in the periosteum, that at these later stages, tdTomato cells are not strongly responding to a Hh signal (*Figure 3C* and *Figure 3—figure supplement 1C*). Furthermore, in cells neighboring Tdtomato+ cells, we did not see any enrichment of *Ptch1* or *Gli2* expression suggesting that they were not receiving a potent Hh signal that could be released by Tdtomato+ cells.

We then investigated the consequence of *Smo* deletion on cartilage callus formation. Surprisingly, *Smo* deletion using either the Pre or Post tamoxifen treatment resulted in a similarly near-complete loss of the cartilage callus at both 7 and 10 dpr (*Figure 4A*, *Figure 4—source data 1*), despite the Pre treatment only deleting *Smo* in a small subset of callus progenitors (*Figures 2C* and *3B*). After *Smo* deletion, although cells did not form a mature callus, they still co-expressed *Sox9* and *Runx2* at 5 dpr (*Figure 4—figure supplement 1A*). At 10 dpr, in contrast to co-expression of *Col2a1* and *Col1a1* throughout the control callus, the few *Col1a1*-expressing cells that formed after *Smo*

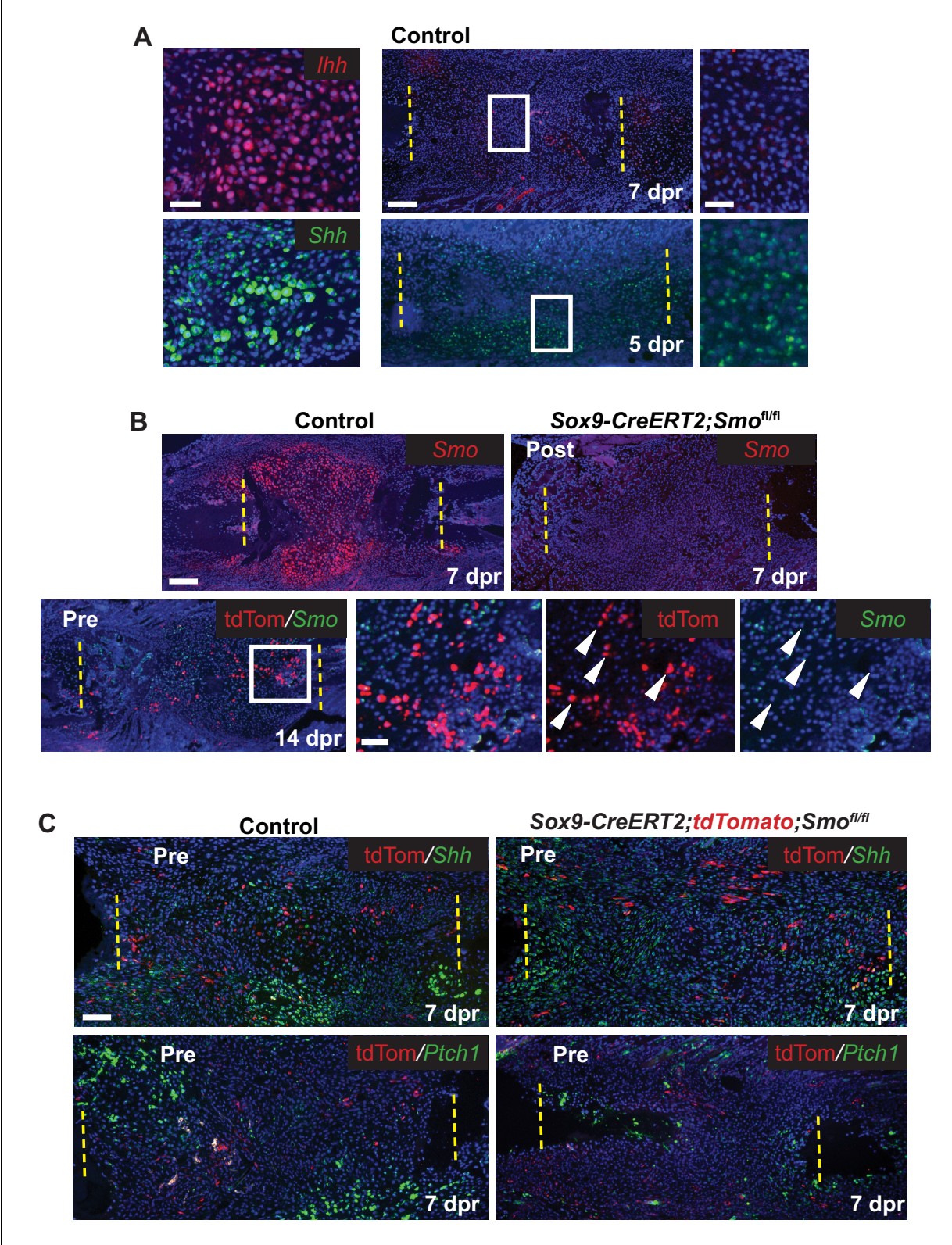

**Figure 3.** Hh signaling during rib repair. (A) Expression of *Ihh* (red) and *Shh* (green) is evident in differentiating cartilage cells. At earlier stages, prior to cartilage formation, *Ihh* is hard to detect even at 7 dpr, while *Shh* is expressed at 5 dpr in many cells across the lesion. (B) Many cells express *Smo* (red) in the control callus at 7 dpr, while in the Post KO, *Smo* expression is not detectable. Fluorescent RNA-ISH for *Smo* (green) combined with IF for tdTomato (red) shows that when using the Pre induction schedule, most Tdtomato+ cells are negative for *Smo* expression, while many non-Sox9-

*Figure 3 continued on next page*

*Figure 3 continued*

positive lineage cells still express *Smo*. White arrow heads indicate tdTomato+ cells that are negative for *Smo* expression. (C) IF for the tdTomato protein (red) in combination with RNA-ISH for *Shh* and *Ptch1* (green) at 7 dpr. In control mice, *Shh* and *Ptch1* expression can be seen in many cells across the callus but most strongly in the chondrocytes. Cells neighboring tdTomato+ cells do not have strong *Ptch1* expression. In Pre KO mice, tdTomato+ cells can be seen throughout the callus. *Ptch1* expression is strongest in the small regions of cartilage that form at the cut ends. Scale bar A = 200 microns, enlarged boxes = 50 microns; B = 200 microns, enlarged boxes = 50 microns; C = 100 microns.

DOI: https://doi.org/10.7554/eLife.40715.010

The following figure supplement is available for figure 3:

**Figure supplement 1.** Characterization of Tdtomato+ cells during repair.

DOI: https://doi.org/10.7554/eLife.40715.011

deletion lacked high levels of *Col2a1* and vice versa, demonstrating a lack of hybrid osteochondral cells (*Figure 3B*).

The lack of callus formation and hybrid cells following *Smo* deletion was not reflected by altered numbers of pHH3+ proliferative or TUNEL+ apoptotic cells at either 7 or 10 dpr, and lineage tracing with the tdTomato reporter revealed similar numbers of cells within the resection site at 10 dpr (*Figure 4C*, *Figure 4—source data 2*, *Figure 4D*, *Figure 4—source data 3*, *Figure 4—figure supplement 1B*, *Figure 4*-figure supplement-source data 1). Thus, Hh signaling is likely not required for the early proliferative expansion in response to injury or for cell survival. Instead, we conclude that Hh signaling is required in Sox9+ cells for promoting the differentiation of non-Sox9 lineage cells into hybrid osteochondral cells of the callus. When the Sox9+ Pre population is null for *Smo*, non-Sox9 lineage cells are still unable to differentiate (despite expressing *Ptch1*) resulting in the near complete failure of callus formation. Thus, this minority Sox9+ periosteal lineage population plays a critical initiating role in callus differentiation.

## Requirement of Hh signaling in Sox9+ lineage cells for rib bone regeneration

We next assessed the consequence of defective callus formation on subsequent regeneration of rib bone in *Smo*-deleted animals. In contrast to control animals showing robust *Col1a1* and *Col2a1* co-expression in cells lining new trabecular bone at 14 dpr, deletion of *Smo* using either the Pre or Post regimens resulted in a near complete absence of co-expressing cells. Instead, only small numbers of *Col2a1*-only cells were observed, primarily near the cut ends of the bone, while cells with osteo-blast morphology expressed predominantly only *Col1a1* (*Figure 5A*). Analysis of H and E-stained histological sections confirmed a marked decrease in bone formation at 14 dpr, despite substantial mesenchyme still observed in the resection site, with the magnitude of the bone defect similar in both regimens (*Figure 5B*). We found that 'Late' removal of *Smo* (injection of tamoxifen at 3–5 dpr targeting the whole callus) only caused a slight delay in repair with a fully bridged callus evident at 14 dpr (*Figure 5—figure supplement 1*) suggesting that the decrease in bone formation seen in both the Pre and Post regimens is largely related to reduced cartilage callus formation. Whole-mount staining with alizarin red and alcian blue confirmed a failure of bone union at 4 and 6 wpr in both Pre and Post conditions (*Figure 5C*). These findings demonstrate that the Sox9+ subpopulation requires Hh signaling not only to build the repair callus but that a substantial cartilage callus may be needed to efficiently regenerate bone.

## Discussion

We find that building an extensive callus of a unique type of hybrid osteochondral skeletal cell is essential for successfully bridging large gaps in adult mammalian rib bone. During this process, Hh signaling plays a critical role, distinct from that in the developing growth plate, in promoting the ultimate differentiation of these hybrid osteochondral skeletal cells. Further, we provide genetic evidence that Hh signaling acts upon a rare periosteal subpopulation of *Sox9*-expressing cells, that behave as messenger cells. These Sox9+ periosteal cells then stimulate, through a yet to be determined signal, their neighboring non-Sox9-lineage cells, which constitute a majority of the callus, to differentiate and build new callus and bone (summarized in *Figure 6*).

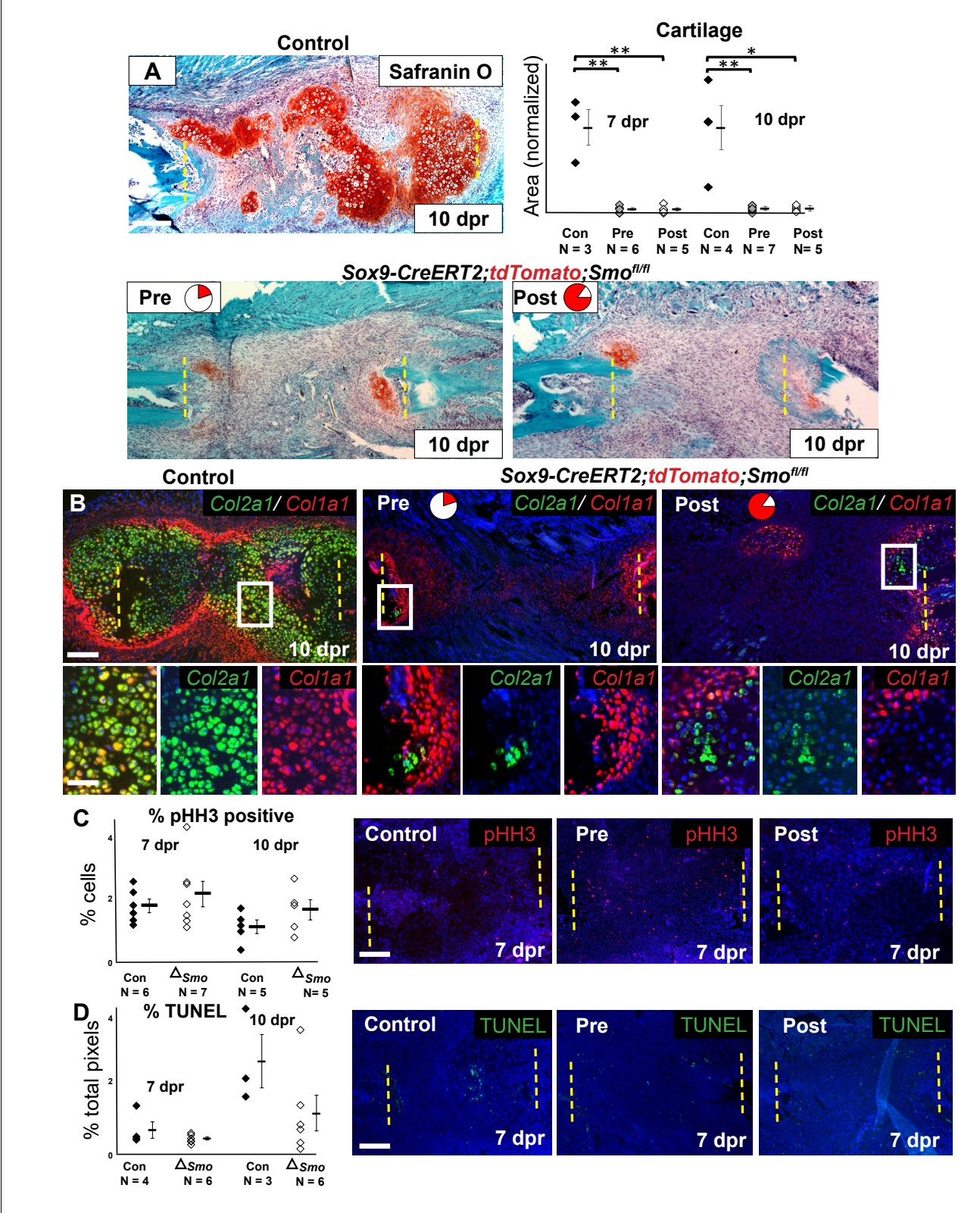

**Figure 4.** Requirement of Hh signaling for rib callus formation. (A) Safranin O staining was used to visualize cartilage formation. At 10 dpr, the control callus has significantly more cartilage then both the Pre KO and Post KO repair calluses. In the graph showing the quantification of cartilage based on Safranin O staining at both 7 and 10 dpr, data are presented as compared to the average of controls, normalized to 1. When comparing both the Pre and Post KO to the control, the difference in the amount of cartilage is statistically significant, but between the Pre and Post KO, there is no statistically

*Figure 4 continued on next page*

*Figure 4 continued*

significant difference. See *Figure 4—source data 1*. (B) Double fluorescent RNA-ISH of *Col1a1* (red) and *Col2a1* (green) expression shows that there are fewer hybrid cells that mediate large scale repair in the KO calluses when compared to the control. Most of the cells in the control callus express high levels of both *Col1a1* and *Col2a1*, while in both Pre and Post KO calluses many of the chondrocytes only express *Col2a1*. The enlarged boxes show cells with chondrocyte morphology with color channels merged and separated. (C) IF against pHH3 was used to mark cells undergoing proliferation. The percentage of positive cells in the callus was calculated vs. the total number of callus cells. No statistically significant difference between the control (n = 6, 5) and the *Sox9-CreERT2; Smo*$^{fl/fl}$ (n = 7, 5) mice at 7 or 10 dpr (p=0.460 and 0.210 respectively) was evident. Representative panels are shown. See *Figure 4—source data 2*. (D) TUNEL staining was used to detect apoptotic cell death at 7 and 10 dpr. The graph shows the percentage of green pixels, in comparison to total pixels in the callus area. There is no statistically significant difference between the control (n = 4,3) and *Sox9-creERT2;Smo*$^{fl/fl}$ (n = 6,6) mice at 7 and 10 dpr (p=0.243 and 0.141 respectively). Representative panels are shown. See *Figure 4—source data 3*. Whisker bars are mean ± SEM. Statistical differences were determined using the unpaired *t* test. *p value < 0.005 **p value < 0.001; Scale bar A = 200 microns, B = 50 microns, enlarged box = 25 microns, C, D = 200 microns.

DOI: https://doi.org/10.7554/eLife.40715.012

The following source data and figure supplements are available for figure 4:

**Source data 1.** Quantification of Safranin O.
DOI: https://doi.org/10.7554/eLife.40715.015
**Source data 2.** Quantification of pHH3.
DOI: https://doi.org/10.7554/eLife.40715.016
**Source data 3.** Quantification of TUNEL positivity.
DOI: https://doi.org/10.7554/eLife.40715.017
**Figure supplement 1.** Characterization of *Smo* knock-out calluses.
DOI: https://doi.org/10.7554/eLife.40715.013
**Figure supplement 1—source data 1.** Quantification of cells that are tdTomato+.
DOI: https://doi.org/10.7554/eLife.40715.014

We observe high-level co-expression of genes typically associated with cartilage (*Sox9*, *Col2a1*, *Col10a1*) or bone (*Col1a1*, *Bglap*) in callus cells during murine rib bone regeneration. This is in marked contrast to the largely exclusive expression of cartilage versus bone genes in the developing growth plates, although hypertrophic chondrocytes do express low levels of many bone-associated genes (*Gerstenfeld and Shapiro, 1996*). The trajectory of hybrid cells is also fundamentally different from the subpopulation that is proposed to 'transdifferentiate' from chondrocytes to osteoblasts in the growth plate (*Bahney et al., 2014*; *Shimomura et al., 1975*; *von der Mark and von der Mark, 1977*; *Yang et al., 2014*; *Zhou et al., 2014*). In the growth plate, these transdifferentiating cells express chondrocyte-associated genes first and then, through a process that remains mysterious, turn off chondrocyte-associated genes and turn on osteoblast-associated genes. Here, we confirm a similar process of 'first cartilage and then bone' in the rib growth plate. In contrast, in the regenerating rib, we observe that callus cells co-express cartilage- and bone-associated genes at the earliest stages. We propose that they then maintain this hybrid osteochondral identity as they shift from making a cartilaginous and then a bone-like matrix. We also observe a similarly hybrid osteochondral cell in the femur fracture callus, although further investigations will be required to determine how similar these cells are between the rib and femur callus. Moreover, similar hybrid cells have been reported during zebrafish jawbone repair (*Paul et al., 2016*) suggesting that skeletal cells with dual chondrocyte/osteoblast properties may be critical for bone regeneration across vertebrate species.

Co-expression of bone and cartilage programs in the same cell is certainly not unique to the regenerating callus. For example, *Col1a1* and *Col2a1* are also highly expressed in fibrocartilage cells, however this tissue is morphologically quite different in terms of its high fiber to cell ratio when compared to the regenerating rib callus (*Benjamin and Ralphs, 2004*). Of note, a rare developmental skeletal type has been described, historically referred to as 'chondroid bone' or 'secondary cartilage', that does share many features with the hybrid osteochondral cells we observe during regeneration (*Goret-Nicaise, 1984*; *Shibata and Yokohama-Tamaki, 2008*). Therefore, it may be that these cells are not unique to regeneration, but rather are selectively utilized due to their ability to rapidly proliferate in a relatively avascular environment (a property of cartilage [*Dennis et al., 2015*]) while directly producing mineralized matrix (a property of bone).

Whereas some bone does form in our *Smo*-deleted mice, this is not sufficient to bridge the lesion and healing fails, resulting in a persistent non-union. Cells within this bone do not display hybrid

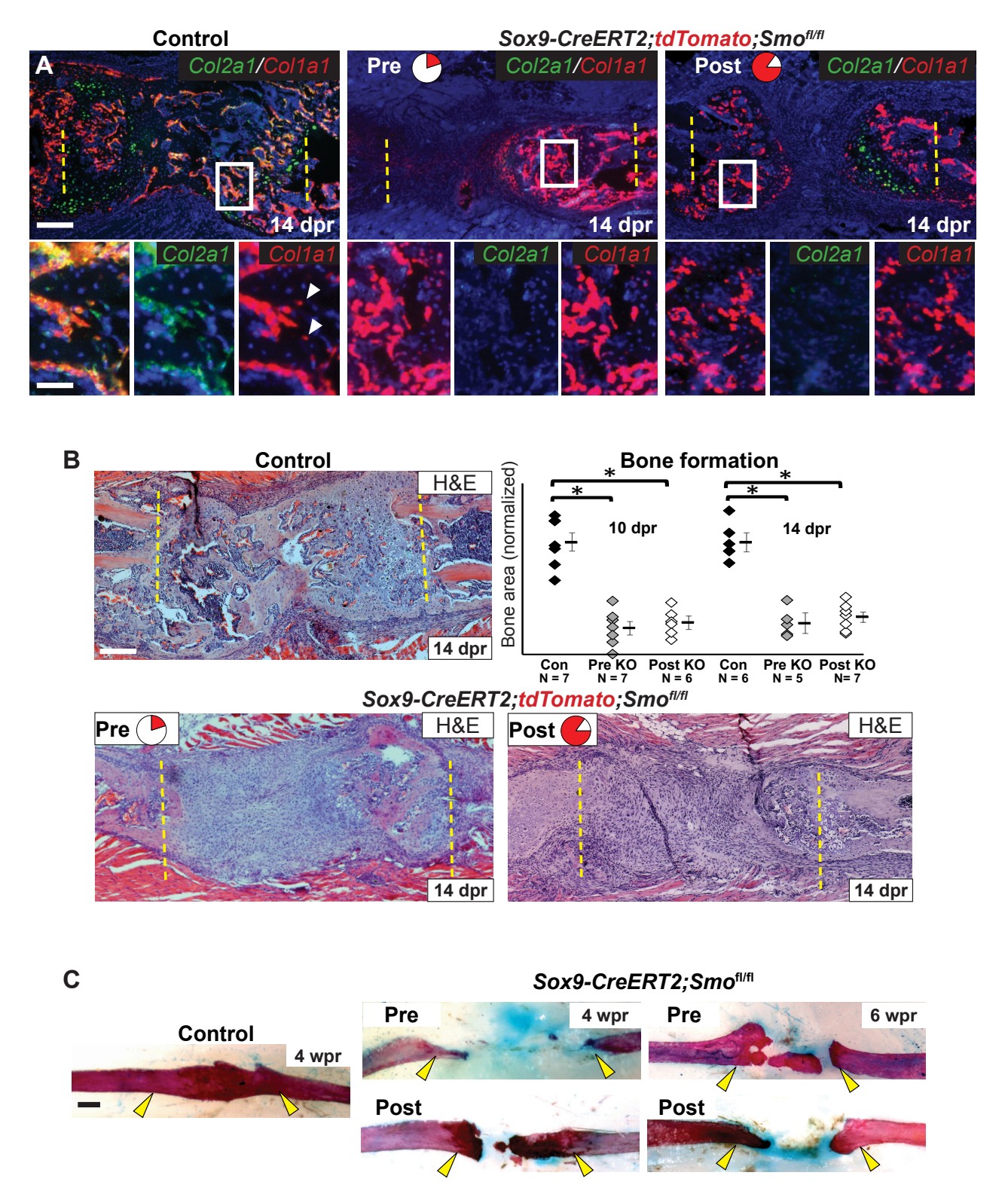

**Figure 5.** Hh signaling is required for bone formation. (**A**) Double fluorescent RNA-ISH of *Col1a1* (red) and *Col2a1* (green). The enlarged boxes show an area of newly formed trabecular bone in separate channels. In both KO contexts, the cells building the trabecular bone express *Col1a1* at high levels, but are largely negative for the expression of *Col2a1*. While, in the control, cells lining the trabecular bone express high levels of both. White arrowheads point to differentiating osteocytes that still express *Cola1*. (**B**) H and E staining at 14 dpr shows the histology of the repair callus in control,

*Figure 5 continued on next page*

*Figure 5 continued*

Pre, and Post KO animals. Both KO mice have much less bone than in controls and many of the cells that have entered the lesion have a progenitor-like morphology. Bone formation was quantified based on histology and the data is shown compared to the average of controls which has been normalized to one in the graph. At both 10 and 14 dpr, both the Pre KO and the Post KO have significantly less bone when compared to the controls. No statistically significant difference is seen when comparing the Pre to the Post KO. See *Figure 5—source data 1*. Whiskers show mean ± SEM. Statistical differences were determined using the unpaired *t* test. *p value < 0.0002 (C) Alizarin red and alcian blue whole mount staining show that at 4 wpr, the resected region is fully spanned by mineralized material in control mice (n = 3), while both the Pre KO and Post KO animals fail to heal. Similar results can still be seen at 6 wpr. Scale bar A, B = 200 microns, enlarged box = 50 microns, C = 500 microns.

DOI: https://doi.org/10.7554/eLife.40715.018

The following source data and figure supplement are available for figure 5:

**Source data 1.** Quantification of bone.
DOI: https://doi.org/10.7554/eLife.40715.020
**Figure supplement 1.** Late KO of *Smo*.
DOI: https://doi.org/10.7554/eLife.40715.019

chondrocyte/osteoblast character, consistent with residual bone forming by direct ossification rather than ossification through a callus intermediate. These findings suggest that during large-scale bone regeneration, Hh signaling in *Sox9*-expressing periosteal cells is selectively required for the formation of a hybrid osteochondral callus.

Hh signaling may play an important but more subtle role in osteocyte maturation as fractures still heal in the absence of *Smo* but with decreased or delayed bone formation (*Baht et al., 2014*; *Wang et al., 2010*). Similarly, we see production of bone (although delayed) in our rib resection model with a Late KO of *Smo* (*Figure 5—figure supplement 1*). In contrast to our rib model, however, loss of *Smo* in these femur/tibia fracture assays still resulted in the formation of a cartilage callus. We do not know why these results contrast from ours where cartilage callus formation is dramatically compromised. One possibility is that the efficacy of *Smo* removal in these fracture experiments was not efficient during cartilage callus stages, alternatively, there could be different requirements for Hh signaling depending on the type of injury (resection vs. fracture). Interestingly, the formation of a cartilage callus may not be as critical during fracture repair, as fractures *can* repair solely through direct ossification (*Colnot et al., 2003*). While in contrast, during large-scale repair, the process of direct ossification is not sufficient to build large pieces of bone and instead, building a hybrid osteochondral callus may be particularly important for bridging large bone gaps.

Hh signaling is known to promote chondrocyte proliferation and osteoblast differentiation in the developing growth plate (*Long et al., 2004*; *Long et al., 2001*). Surprisingly, we found that loss of Hh signaling did not affect the early proliferation of callus cells or the differentiation of osteoblasts in the residual directly ossifying bone. Instead, we provide evidence that Hh signaling has a distinct and essential role in promoting the differentiation of *Sox9/Runx2* expressing progenitors into the hybrid osteochondral skeletal cells that form the repair callus. These results indicate that the regeneration of the rib bone, including its dependence on Hh signaling, does not simply the recapitulate developmental processes seen at the growth plate.

While the heterogeneity of the cells in the periosteum and their lineage relationships remain incompletely understood, we propose that Sox9+ periosteal cells or a subpopulation within them, play an essential instructive role for callus formation during large-scale bone regeneration that has not been previously described in the context of repair. The Sox9+ periosteal subpopulation may be distinct from the previously described subpopulations that are Gli1+, αSMA+, Gremlin1+, or Axin2 +, as RNA-seq analysis of the periosteum from an uninjured femur indicated that *Gli1*, *αSMA*, *Axin2*, and *Gremlin1* are not highly expressed in the Sox9+ cells (*He et al., 2017*). Future studies tracing the lineage of these other populations and determining their dependence on Hh signaling will be needed to resolve whether Sox9+ cells represent a subset of one of these other more abundant populations, or alternatively a distinct progenitor subpopulation. In addition, efforts to delineate potential differences that may exist in periosteal populations from different bones, may explain why some bones regenerate well, while others do not.

One of the most striking findings from our study is that Sox9+ cells are essential for efficient callus formation and rib bone regeneration, despite contributing to only a minority of cells within the callus and regenerated bone. Based on this observation, we propose that the Sox9+ cells act as

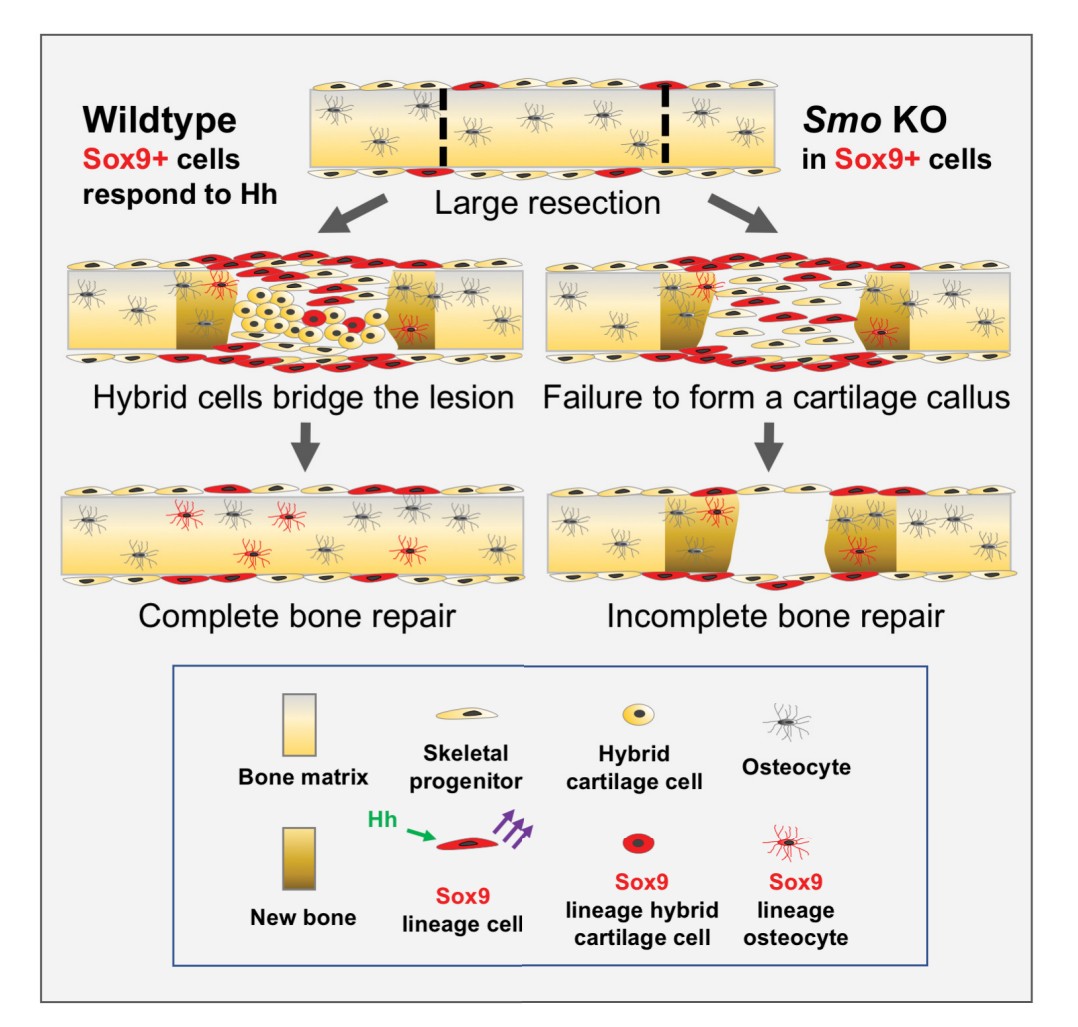

**Figure 6.** Model for large-scale bone repair. In wildtype animals, represented by the left side of the diagram, extensive gaps in the mouse rib (as defined by the hatched lines) can naturally regenerate. A *Sox9*-expressing periosteal subpopulation (indicated in red) along with other skeletal progenitors (yellow) proliferate and migrate into the lesion. These Sox9+ lineage cells require Hh signaling (green arrows) to be able to signal via a yet-to-be identified mechanism to neighboring cells (purple arrows). This signal induces neighboring cells to differentiate into a reparative callus with hybrid osteochondral qualities, leading to complete bridging and bone repair. Sox9 + lineage cells ultimately contribute to the callus and regenerated bone (indicated in brown) although they are represented in the minority. The right side of the diagram represents the outcome of a Pre regimen KO of *Smo* (tamoxifen administered *prior* to injury). When *Smo* is removed from the Sox9+ periosteal subpopulation prior to resection, the Sox9+ lineage cells can still contribute but are not activated and therefore do not relay a differentiation signal to neighboring cells. Thus, the entire callus fails to differentiate into a hybrid osteochondral callus. While some bone forms via direct ossification, this is not sufficient and healing fails.

DOI: https://doi.org/10.7554/eLife.40715.021

'messenger' cells by releasing a yet to be determined signal that promotes the differentiation of neighboring cells Sox9-negative lineage cells. One possibility is that Sox9+ progenitors differentiate early in response to the initial Hh signal (potentially *Shh,* since strong expression is evident early) and then propagate *another* wave of Hh signaling to neighboring Sox9-negative cells, similar to the role of Hh in the morphogenetic furrow during *Drosophila* eye development (*Domínguez and Hafen, 1997*; *Kumar, 2011*; *Ma et al., 1993*). The failure of Sox9+ cells to differentiate and thus express *Shh* and *Ihh,* could then lower the total Hh signal in the callus to below a critical threshold need for cartilage differentiation. However, our results suggest that Sox9+ cells are not the first to

differentiate and while they likely respond to a Hh signal early in the periosteum, they are likely not strong propagators of a further Hh signal as cells nearby Sox9+ lineage cells do not display a strong upregulation of Hh pathway read-outs as the callus differentiates. While a response to Hh signaling may require more sensitive assays, we found the expression of both *Ptch1* and *Gli2* to be very low in undifferentiated cells at 7 dpr in both control and Pre *Smo* KO calluses *Figure 3C* and *Figure 3—figure supplement 1*) suggesting that they were not receiving a potent secondary Hh signal. In addition, the expression of *Shh* was not particularly strong in the tdTomato+ cells at this stage (*Figure 3C*). Thus, we instead favor a hypothesis, that in response to Hh signaling Sox9+ cells emit another to-be-identified relay signal. This signal then orchestrates repair by promoting the differentiation of neighboring cells from a non-Sox9+ lineage into mature matrix-producing hybrid osteochondral skeletal cells that bridge the lesion (*Figure 6*). It is possible that Hh signaling may have an additional role in later stages of repair (osteocyte differentiation), but our study supports a critical role early in large-scale rib repair. Future investigations into factors produced by Sox9-lineage cells that promote callus formation may lead to better strategies of boosting bone repair in other parts of the body that do not heal as effectively.

# Materials and methods

### Key resources table

| Reagent type | Designation | Source/ Reference | Identifiers | Additional information |
|---|---|---|---|---|
| Genetic reagent (*M. musculus*) | B6.Cg-Tg;Col1a1*2.3-GFP[1Rowe/J] | JAX 013134 | MGI: 151234 | |
| Genetic reagent (*M. musculus*) | Tg;Col10a1-mCherry[3Pmay/J] | *Maye et al., 2011* JAX 017465 | MGI: 5428042 | P Maye |
| Genetic reagent (*M. musculus*) | Sox9[tm1(cre/ERT2)Haak] | *Soeda et al., 2010* | MGI: 4867441 | H Akiyama |
| Genetic reagent (*M. musculus*) | B6;129S6-Gt(ROSA)26Sor[tm9(CAG-tdTomato)Hze/J] | JAX 007905 | MGI: 3809524 | |
| Genetic reagent (*M. musculus*) | Smo[tm2Amc/J] | JAX 004526 | | |
| Antibody | Rabbit polyclonal anti-Collagen I | Abcam ab34710 | | 1:250 |
| Antibody | Goat polyclonal anti-Collagen II | Southern Biotech 1320–01 | | 1:200 |
| Antibody | Rabbit polyclonal anti-pHH3 | Millipore 06–570 | | 1:200 |
| Antibody | Chicken polyclonal anti-mCherry | Novus Biological NBP2-25158SS | | 1:200 Also detects tdTomato |
| Antibody | Alexa Fluor 488 Donkey polyclonal anti Goat IgG (H and L) | Abcam ab150129 | | 1:500 |
| Antibody | Alexa Fluor 568 Goat polyclonal anti-Rabbit IgG (H + L) | ThermoFisher A-11011 | | 1:250 |
| Antibody | Alexa Fluor 568 Goat polyclonal anti-Chicken IgY (H and L) | Abcam ab175477 | | 1:500 |
| Antibody | Alexa Fluor 488 Donkey anti-Goat IgG (H and L) | Abcam ab150129 | | 1:500 |

## Mice and animal housing

All procedures were approved by the University of Southern California Institutional Animal Care and Use Committee (Protocol #: 11256, 20639). We used the following mouse lines: *Sox9-CreERT2*

(Sox9[tm1(cre/ERT2)Haak] [*Soeda et al., 2010*]), *R26R-tdTomato* (B6;129S6-*Gt(ROSA)26Sor*[tm9(CAG-tdTo-mato)Hze]/J; JAX 007905), *Smo*[fl/fl] (*Smo*[tm2Amc]/J; JAX 004526, *Col1a1(2.3-GFP)* (B6.Cg-Tg; Col1a1*2.3-GFP)[1Rowe/J];JAX 013134), Col10a1-mCherry (*Maye et al., 2011*). To generate *Sox9-CreERT2;tdTo-mato;Smo*[fl/fl] mice, *Sox9-CreERT2;tdTomato* males were crossed to *Smo*[fl/fl] females and *Sox9-CreERT2;tdTomato;Smo*[fl/+] offspring males were back-crossed to *Smo*[fl/fl] females to generate *Sox9-CreERT2;tdTomato;Smo*[fl/fl] offspring. Both male and female mice between 6–8 weeks old were used for experiments. Control mice were uninduced siblings or tamoxifen-induced *Sox9-CreERT2;tdTo-mato* mice.

## Injury assays

Rib resections were performed as previously described (*Tripuraneni et al., 2015*) with the modifica-tion that a bone segment of 3 mm was removed and that post-operative pain was managed with buprenorphine SR (ZooPharm) at a dose of 0.5 uL/gram. Rib repair was assessed after set healing time points: 0 dpr – 10 wpr. To induce Cre recombination, 100 uL of a 20 mg/ml stock of Tamoxifen (Sigma-Aldrich: T5648-1G, dissolved in corn oil at 60°C for 2 hr) was used per injection. Tamoxifen injections were administered intraperitoneally using a 25-gauge needle. Two injection schedules were used: 1) three consecutive injections starting 7 days before resection ('Pre' regimen), 2) four consecutive injections starting 1 day prior to surgery ('Post' regiment). Uninduced controls were injected with corn oil only. Femur fracture assays were carried out as previously described (*He et al., 2017*).

## Histological analyses

Samples were fixed with 4% PFA overnight at room temperature, decalcified with 20% ETDA at pH 7.5 for 10–14 days, and then processed for paraffin embedding. A microtome (Shandon Finesse Me +: 77500102) was used to cut paraffin sections seven microns thick. The sections were mounted on Superfrost Plus slides (VWR, 48311–703). After deparaffinizing slides in Citrisolv and rehydrating, H and E or Safranin O staining was carried out using standard protocols. Skeletal staining was per-formed on EtOH-fixed samples (*Rigueur and Lyons, 2014*). To visualize native tdTomato fluores-cence, samples were fixed in 4% PFA on ice for 30 min and placed in 30% sucrose overnight at room temperature. The samples were embedded in OCT and flash frozen in an EtOH dry ice bath. 10 µM thick sections were cut using a Leica CM3050 S cryostat. Tape (cryofilm type 3C(16UF) C-FUF303) was used to preserve the histology of the bone (*Kawamoto, 2003*). OCT was removed with a 1xPBS wash before mounting.

## Immunofluorescence

Detection of pHH3 and Tdtomato proteins was carried out on paraffin sections. Slides were de-waxed and cells were permeabilized with 0.1% Triton-X followed by antigen retrieval in 10 mM sodium citrate, 0.05% Tween 20, pH of 6.0 in a 95°C water bath for 30 min. Slides were blocked in 20% serum for 1 hr and then incubated with the primary antibodies overnight at 4°C (anti-pHH3, Millipore: 06–570, 1:200; anti-mCherry which also detects tdTomato, Novus Biological: NBP2-25158SS, 1:200; anti-Col1, Abcam: ab34710, 1:250; anti-Col2, Southern Biotech: 1320–01, 1:200). Secondary antibodies used were: Alexa Fluor 568 goat anti-rabbit (ThermoFisher: A-11011, 1:250), Alexa Fluor 568 goat anti-chicken (Abcam: ab175477, 1:500), and Alexa Fluor 488 donkey anti-goat (Abcam:ab150129, 1:500).

## TUNEL

To detect apoptosis, the In Situ Cell Death Detection Kit, Fluorescein (Sigma-Aldrich: 11684795910) was used as directed.

## RNA in situ hybridization

Fluorescent and colorimetric RNA in situ hybridization (RNA-ISH) was performed on 7 µm paraffin sections as previously described (*Paul et al., 2016*). Complementary DIG or FL labeled RNA probes were generated following kit instructions (Sigma-Aldrich: 11277073910 and 11685619910) and were detected with Anti-Digoxigenin-POD (Sigma-Aldrich: 11207733910) and Anti-Fluorescein-POD (Sigma-Aldrich: 11426346910). For double fluorescent RNA-ISH the TSA Cyanine three and

Fluorescein system from Perkin Elmer was used as directed (NEL753001KT). For colorimetric RNA-ISH Anti-Digoxigenin-AP (Sigma-Aldrich: 11093274910) was used to detect the probes.

Probes were generated to the following sequences:

| Gene | Nucleotides | Sequence ID |
| --- | --- | --- |
| Col2a1 | 2166–3020 | XM 006520386.6 |
| Sox9 | 1144–1398 | NM 011448.4 |
| Col1a1 | 3563–4475 | NM 007742.4 |
| Bglap | 39–342 | NM 001032298.3 |
| Ihh | 1110–1684 | NM 010544.3 |
| Ptch1 | 134–750 | XM 006517163.3 |
| Smo | 373–745 | NM 176996.4 |
| Runx2 | 17–285 | NM 009820.5 |
| Col10a1 | 1624–2114 | NM 009925.4 |

Slides with fluorescence were mounted with Vectashield with DAPI (Vector Laboratories: H1200) and were imaged with a Nikon AZ100 Macroscope and photographed (Nikon Digital sight DS-Fi1). Fluorescent images were edited for contrast and color levels in Adobe Photoshop CS5.

## Quantification and statistical analysis

For quantification, all images were taken at the same magnification for each data set. Student's t-test was used to compare groups. A probability value of 0.05 or less was marked as significant. Each data point was plotted on the scatter plot and the mean was defined on the graph, unless the data set was normalized. Statistical tests were performed using GraphPad.

To determine the amount of cartilage, a mid-sagittal section through both mutant and control sample was stained with Safranin O and quantified in ImageJ. In brief, the image was thresholded for the Safranin O color (orange) and the area of the color was measured. To determine bone area, samples were stained with H and E and analyzed with the BioQuant image analysis program. The bone area was compared to the entire resected area. The values were normalized within each time point.

Quantification of the number of cells expressing pHH3 was carried out using the analyze particle function in ImageJ with the repair callus defined as the area of interest. The red channel was used to count the number of cells positive for pHH3 while the blue channel was used to count the total number of cells (nuclei stained with DAPI). The ratio of pHH3 positive cells to total cells was used to calculate the percentage of cells in mitosis. This method was also used to quantify the number of tdTomato+ cells within the callus. To quantify apoptosis, Adobe Photoshop CS5 was used to analyze the green channel. The number of green pixels compared to the total number of blue pixels (DAPI) in the region of interest was calculated. Analysis of pHH3 and TUNEL positivity was done on de-identified images by several laboratory members.

## Acknowledgements

We thank Andy P McMahon, Yang Chai, and Baruch Frenkel for advice and comments on the manuscript, Xinjun He for sharing results before publication, Drs. H Akiyama and P Maye for sharing mouse lines, and Ashlie Muñoz, Simone Schindler, Sophia Bougioukli for technical assistance. Funding was from the National Institutes of Health [T32 HD060549 to STK, R21 DE023899 and R35 DE027550 to JGC; R21 AR064462 and R01 AR069700 to FVM]; the James H Zumberge Research and Innovation Fund to FVM; and a University of Southern California Regenerative Medicine Initiative Award to FVM, JGC, and JRL.

# Additional information

## Funding

| Funder | Grant reference number | Author |
|---|---|---|
| National Institute of Arthritis and Musculoskeletal and Skin Diseases | R21 AR064462 | Stephanie T Kuwahara<br>Maxwell A Serowoky<br>Nikita Tripuraneni<br>Neel V Hegde |
| Eunice Kennedy Shriver National Institute of Child Health and Human Development | T32 HD060549 | Stephanie T Kuwahara |
| University of Southern California | USC Regenerative Medicine Initiative | Stephanie T Kuwahara<br>Nikita Tripuraneni<br>Jay R Lieberman<br>J Gage Crump<br>Francesca V Mariani |
| National Institute of Arthritis and Musculoskeletal and Skin Diseases | R01 AR069700 | Stephanie T Kuwahara<br>Maxwell A Serowoky<br>Nikita Tripuraneni<br>Neel V Hegde<br>Jay R Lieberman |
| National Institute of Dental and Craniofacial Research | DE023899 | J Gage Crump |
| National Institute of Dental and Craniofacial Research | R35 DE027550 | J Gage Crump |
| University of Southern California | Zumberge Innovation Award | Francesca V Mariani |

The funders had no role in study design, data collection and interpretation, or the decision to submit the work for publication.

## Author contributions

Stephanie T Kuwahara, Conceptualization, Data curation, Formal analysis, Validation, Investigation, Visualization, Methodology, Writing—original draft, Writing—review and editing; Maxwell A Serowoky, Data curation, Formal analysis, Investigation, Writing—review and editing; Venus Vakhshori, Investigation, Methodology, Femur fracture generation; Nikita Tripuraneni, Conceptualization, Investigation, Methodology; Neel V Hegde, Data curation, Formal analysis, Investigation; Jay R Lieberman, Conceptualization, Supervision, Writing—review and editing; J Gage Crump, Conceptualization, Funding acquisition, Methodology, Writing—original draft, Project administration, Writing—review and editing; Francesca V Mariani, Conceptualization, Supervision, Funding acquisition, Investigation, Methodology, Writing—original draft, Project administration, Writing—review and editing

## Author ORCIDs

Maxwell A Serowoky (iD) http://orcid.org/0000-0001-9361-3997
J Gage Crump (iD) https://orcid.org/0000-0002-3209-0026
Francesca V Mariani (iD) https://orcid.org/0000-0003-1619-8763

## Ethics

Animal experimentation: This study was performed in strict accordance with the recommendations in the Guide for the Care and Use of Laboratory Animals of the National Institutes of Health. All of the animals were handled according to approved institutional animal care and use committee (IACUC) protocols (#11256 and #20639) of the University of Southern California.

## Decision letter and Author response

Decision letter https://doi.org/10.7554/eLife.40715.024

Author response https://doi.org/10.7554/eLife.40715.025

---

## Additional files

### Supplementary files

• Transparent reporting form
DOI: https://doi.org/10.7554/eLife.40715.023

### Data availability

All data generated or analysed during this study are included in the manuscript and supporting files.

---

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
