## [Decision Letter]

[**Editorial note:** This article has been through an editorial process in which the authors decide how to respond to the issues raised during peer review. The Reviewing Editor's assessment is that all the issues have been addressed.]

Evaluation of resubmission:

The authors have addressed our concerns during the re-revision. Over time, the focus of the paper has changed considerably; the authors now also suggest there may exist secondary signals from the Sox9 lineage cells that are not Hh which are critical in this process of large-scale repair. Another concern we had was regarding Figure 2C, which has also been addressed. The quality of the data is now convincing.

Decision letter after peer review:

Thank you for submitting your article "Pioneer cells orchestrate large-scale skeletal regeneration in mammals" for consideration by *eLife*. Your article has been reviewed by three peer reviewers, and the evaluation has been overseen by Marianne Bronner as the Senior Editor and Reviewing Editor. The reviewers have opted to remain anonymous.

The Reviewing Editor has highlighted the concerns that require revision and/or responses, and we have included the separate reviews below for your consideration. If you have any questions, please do not hesitate to contact us.

Summary:

In this paper, Kuwahara and colleagues investigate the regeneration of rib bones in mice, using a model of rib ablation. They characterized the population of cells, all derived from the periosteum, which repair the resected rib. They find that in the repairing callus, cells co-express chondrocyte-specific and osteoblast-specific markers. Using lineage tracing techniques and *Sox9* as a marker of the chondrocyte lineage, they find that the callus comprises a minor (1/3) population derived from cells expressing *Sox9* in the periosteum, while the majority (2/3) upregulate it after resection. The ablation of Smo in the minor population or in the entire population leads to a similar phenotype, defects in callus formation and bone differentiation. The authors conclude that the periosteum-derived *Sox9* cells play an essential function in the HH-mediated bone regeneration of ribs.

Major concerns:

1) The main concern with the manuscript is that a mechanistic understanding of the phenotype obtained after deletion of Smo in the two experimental paradigms is lacking. How is it that the arrest of HH signalling in a portion of the cells that repair the rib impacts the entire differentiation program of this population?

2) Is Hh signaling uniquely involved in regeneration or simply necessary in general for ossification?

Separate reviews;

*Reviewer #1:*

In this work, Kuwahara et al., examine rib regeneration using a murine model. They show that the rib regenerates from cells located in the periosteum and that Hedgehog signaling is required for the initial differentiation of the cells that will comprise the regenerating tissue. These cells are derived from a population of cells that express *Sox9*, and the appear to give rise to cells that co-express *Col1* and *Col2*. They propose a model whereby a rare population of *Sox9*+ cells in the periosteum recruit other cells to participate in the regenerative process by making the "hybrid cells" that co-express *Col1* and *Col2*. The major finding of this work are two-fold. First, what appear to be chondorcytes co-express *Col1* and *Col2*-markers of osteoblasts and chondrocytes respectively. 2nd that deleting Smoothened from a *Sox9*+ population of cells in the periosteum leads to poor healing and indicating that hedgehog signaling is required at an early stage of cell differentiation. While this is interesting some fo the conclusions are not supported by the experimental approach and resulting data, and there are some concerns about the exclusive use of in situ hybridization to demonstrate the presence of hybrid cells.

I disagree that mammals exhibit a limited capacity for bone repair. In fact, bone is one tissue that exhibits excellent regenerative potential in mammals. The entire basis of distraction osteogenesis is based on the ability of bone to regenerate itself. Further, there is no comparison of healing of the rib to any other bone in the body in this work. So saying the ribs are a "notable exception" is not necessarily correct.

The statement that not all cells in the callus are derived from *Sox9*-expressing progenitor cells is not supported in the literature (He et al., 2017). In fact, Figure 2E shows that by extending the labeling protocol to 2 days after injury, there is a large contribution of *Sox9*+ cells to the regenerating tissue.

Hybrid cells:

It would be interesting to assess rib healing when the periosteum is removed along with the bone. Do hybrid cells still appear?

Why is there no *Col1* expression in the cells embedded in the new trabecular bone?

Do the hybrid cells co-express *Col1* and *Col10*? Or just *Col1* and *Col2*? It is hard to assess this from the figures.

Hedgehog experiments:

Shh expression has been examined in other work on rib fracture models (Matsumoto et al., 2016), and it appears the results are different from those here. This should be discussed.

Where is Shh, Ihh, Ptc expressed prior to chondrocyte differentiation? It seems this is a key piece of data that is missing. Instead Figure 3 shows expression of these genes in the chondrocytes. However, knock-out of Smoothened affects an earlier stage of healing. Work by Baht has already shown that Hedgehog signaling is dispensable during expansion of the cartilage in the callus. Thus, there is a disconnect between the expression patterns of Shh, Ihh, Smo, and Ptc, and the experimental data.

Model:

In Figure 5, I do not see evidence in the manuscript of *Sox9*+ osteocytes or hybrid cartilage cells that do not express *Sox9*. Also, it is not clear why there are *Sox9*- hybrid cells since when lineage tracing was extended to after the injury, nearly all of the regenerated tissue was derived from labeled cells (Figure 2E). It seems that many more cells express *Sox9* after injury, which could just reflect activation of a larger number of the stem cell pool by the injury itself, and not reflect a recruitment of these cells by the prior *Sox9*+ population as discussed in paragraph one of the Discussion section. To show that these cells act as pioneers that recruit other cells, the *Sox9*+ cells would need to be depleted prior to injury and healing examined.

Minor Comments:

Figures 2 and Figure 2—figure supplement 2 should be combined. It is difficult to compare the location of the images when the histology and in situs are in separate figures.

Additional data files and statistical comments:

An additional method of assessing co-expression of *Col1* and *Col2* should be considered, either PCR after FACS, immunohistochemistry, or an alternate method. Since this is a central tenet in the paper, a more robust analysis would be useful.

*Reviewer #2:*

It is known that rib bones have a higher capacity for repair than most bones, but the basis for this regenerative capacity is unknown. The authors investigate this process by analyzing the repair of rib lesions in adult mice. They propose that *SOX9* expressing cells in the periosteum are important during skeletal regeneration post injury. During this process, cells attain a characteristic intermediate "hybrid state" in the callus, wherein they express markers for both nascent/mature cartilage, as well as differentiated osteoblasts. This is in contrast to the typical process of endochondral ossification, where specific markers are expressed in discrete populations of cartilage with minimal overlap. The authors moreover implicate hedgehog signaling in playing a critical role during formation of the intermediate callus structure during regeneration. They speculate that Hh regulates the differentiation of hybrid cells and eventually leads to large-scale regeneration of these bones.

The analysis presented is detailed and benefits from the use of multiple markers. At the same time, the general message of the work is unclear except to present the idea that cells in the callus are in a "hybrid state" and as such it is not obvious that the manuscript is well suited for a general biology audience.

I have the following specific comments:

1) During endochondral ossification in the growth plate, *SOX9* levels are known to vary in distinct cartilage cell types at various stages of differentiation (Leung et al., 2011, PMID 22072985). Therefore, the authors need to justify that all *Sox9*+ cells of the periosteum are the pioneer population, or they would have to use another marker (and another Cre line, see below).

2) The authors are using a Sox9CreERT2 line to lineage trace the source of cells that give rise to the callus. It is unclear from the data, if most cells are recruited from the region neighboring the injury site or from more distant cells.

3) Other periosteum/osteoblasts specific CreERT2 lines should have been used to validate the periosteum as the source of cells in the callus.

4) The authors mention in their opening statement that "most bones in mammals display limited capacity for repair". Is it known if there is also a reduced number of *Sox9*+ cells in other bones which have limited repair capacity?

5) Does hedgehog signal in an autocrine rather than paracrine manner? It would be good to know if only a subset of cells activate Gli or specific Hh target genes.

6) This reviewer was not convinced by the studies implicating Hh signaling in regeneration. Although the authors observe that disruption of Smo leads to a reduction in *Col2a1/Col1a1* expression in the callus, hedgehog signaling cannot have a unique role in regeneration since it also has a very general role in endochondral ossification.

*Reviewer #3:*

In this paper, Kuwahara et al. investigate the regeneration of rib bones in mice, using a model of rib ablation. They characterized the population of cells, all derived from the periosteum, which repair the resected rib. Using ISH they found, within the repairing callus, cells (which they name "hybrid" skeletal cells) that co-express chondrocyte-specific and osteoblast-specific markers. Using lineage tracing techniques and *Sox9* as a marker of the chondrocyte lineage, they find that the callus comprises a minor (1/3) population derived from cells expressing *Sox9* in the periosteum, while the majority (2/3) upregulate it after resection. The ablation of Smo in the minor population or in the entire population leads to a similar phenotype, defects in callus formation and bone differentiation. The authors conclude that the periosteum-derived *Sox9* cells play an essential function in the HH-mediated bone regeneration of ribs.

In general, the study is interesting and it could be worth publishing in a good journal. The illustrations are of excellent quality, even though a little more infos and enlargements would sometimes be helpful. Below are some comments about the text and the terminology used. In addition, there are a few technical points that could rather easily be addressed. The main concern with the manuscript is that a mechanistic understanding of the phenotype obtained after deletion of Smo in the two experimental paradigms is lacking. How is it that the arrest of HH signalling in a portion of the cells that repair the rib impacts the entire differentiation program of this population? Without this, the study seems a little preliminary.

Text and terminology:

Why the question that is addressed here is worth studying and what are the key findings of the study could be explained more clearly in the Introduction. For non-specialists it takes a lot of reading of this and many other papers (and there are many) to understand what is novel about this study.

The terms "pioneer" and "hybrid" are odd.

Pioneer: that is first to explore or to do something novel: Darwin and Mendel, certainly; pioneer neurons, pioneer muscle cells, maybe. But here it does not seem justified.

Hybrid: characterizes the progeny of two distinct breeds, species. Wouldn't it be more correct to call them bi-potent progenitors?

"Using the tdTomato reporter, we confirmed that the Pre tamoxifen treatment resulted in deletion of Smo in tdTomato+ cells within the resection site. In contrast, the Post tamoxifen treatment resulted in deletion of Smo throughout the resection site (Figure 3—figure supplement 1C)"

These phrases are unclear: the authors oppose "within" and "throughout". What is the difference? Explain better.

Pictures:

Enlarged pictures (maybe insets) of the Figures 1B and C with indications to identify the mesenchymal-like and cartilage-like cells would help non-specialists.

Technical improvements:

Some ISH data are a little confusing. First, they are done at different times: 5 dpr in Figure 1 and 7 dpr in Figure 1—figure supplement 1. Adding to the confusion is the fact that the lineage analyses are started at 10 dpr, such that it is difficult to compare ISH and lineage analyses. Second, the *Sox9* fluorescent ISH indicates that just about all cells of the callus are positive (Figure 1—figure supplement 1A), while a small proportion of them are positive using a colorimetric approach (Figure 1—figure supplement 2). Could you comment this discrepancy?

"To understand whether cells also upregulate *Sox9* expression during callus formation and to also potentially capture more cells within the callus, we instead applied tamoxifen at the time 18 of injury plus 2 days following".

This seems to be an obscure reason to do this experiment. It would be more appropriate to justify the Post- regimen experiment by doing and ISH at 10 dpr and determine whether there is a discrepancy between the number of *Sox9*-derived cells after the Pre- regimen and the number of *Sox9*+ cells as determined by ISH. According to the ISH shown in Figure 1 and Figure 1—figure supplement 2, there should be a major difference between the two which would justify that the authors utilise another approach.

Mechanism:

Since it is clear from the presented data that HH signalling is implicated in callus formation and in the downstream differentiation, the authors need to find an explanation for the similarity between the phenotype obtained after Pre- or Post- Smo ablation. At this stage, such mechanism is lacking.

Are the Smo+ cells still present after the Pre- protocol the same ones (2/3) than the ones that activated *Sox9* after resection? If yes, this would indicate that HH signalling is not implicated in the recruitment of *Sox9* positive cells by the periosteum-derived population.

The presence of Smo does not mean that the pathway is active. To check this, the authors should look at Ptc (as a sign of a response; maybe Gli could be used as well) and HH expression (to determine whether the signal is there or not). If Ptc and HH are normal, then they should find another reason explaining this.

[Editors' note: further revisions were suggested, as described below.]

Thank you for resubmitting your work entitled "*Sox9*+ messenger cells orchestrate large-scale skeletal regeneration in the mammalian rib" for further consideration at *eLife*. Your revised article has been favorably evaluated by Marianne Bronner (Senior Editor), a Reviewing Editor, and three reviewers.

The manuscript has been improved but there are some remaining issues that need to be addressed before acceptance. Specifically, the analysis of the Hh pathway is incomplete and this is a weakness of the paper. The authors have still not addressed the fact that the majority of the cells that co-express *Col2a1* and *Col1a1* are not *Sox9* lineage. This is really important for the entire premise of the paper. In addition, there are other smaller points raised by the reviewers that you can see in the full reviews included below.

*Reviewer #1:*

This work still lacks an appropriate analysis of the Hh pathway at the time that the *Sox9*+ cells are initially stimulated in the periosteum. Since this is the model that is being put forth, it is necessary to show how/where this pathway is working at this early time point. While this was pointed out previously, the authors have done a very thorough job addressing the other concerns.

Minor comments:

It is confusing why this sentence: "Instead, *Sox9*-positive lineage cells require Hh signaling to stimulate neighboring cells to differentiation [sic-typo] into a skeletal cell type with dual properties of chondrocytes and osteoblasts" remains in the Abstract, since in the response to reviewers the authors state: "we conclude that *Sox9*+ cells are not likely to be emitting a second wave of Hh signaling.…" This seems at odds and could be made more clear.

In subsection “Lineage tracing of *Sox9*-expressing cells to the regenerating rib callus”, the statement that "cells within the femur fracture callus expressed *Col1a1* and *Col2a1*" needs to be modified to "some cells," because from the image it appears the vast majority of cells express one or the other of these genes, but not both. There are no negative controls shown to determine levels of background fluorescence.

*Reviewer #2:*

The authors have improved the manuscript, however I still have a major concern that remains unaddressed. If we compare 14 dpr PRE-induction schedule results in Figure 5A CONTROL and Figure 2C (which determine the contribution of *Sox9* lineage at an early 14 dpr stage), then it seems that the majority of the cells that co-express *Col2a1* and *Col1a1* are not *Sox9* lineage. This same concern I had pointed out earlier when I earlier said "cells are recruited from the region neighboring the injury site or from more distant cells". This is not been clarified in the discussion in the current version of the manuscript. This has further consequences if we now address the role of Hh signaling (compare Figure 5 PRE- and POST-induction schedule) in large-scale repair. In both PRE- and POST-induction schedule Smo mutants, none of the cells co-express *Col2a1* and *Col1a1*; this suggests that the major contribution is from an altogether different source/ or an extended periosteum which was laid down post trauma/ etc. This point also needs attention.

To address this issue, the authors might want to revisit the *Sox9CreERT2* lineage tracing. The percentage contribution can be determined for 10, 14 and 21 dpr for both PRE- and POST- induction schedule. In Figure 2C, there seems to be higher tdTomato+ cells in the regenerated region at 10 dpr that 14 dpr (magnification in 21 does not match the others.)

*Reviewer #3:*

The authors have satisfactorily addressed the comments that were previously made. They have performed additional experiments and have re-written portions of the manuscript to accommodate my and other's comments. As a consequence, the quality of the study and of its conclusions have improved, which should be the purpose of the reviewing process. In my view, this study is now ready to be published in *eLife*.

---

## [Author Response]

Major concerns:1) The main concern with the manuscript is that a mechanistic understanding of the phenotype obtained after deletion of Smo in the two experimental paradigms is lacking. How is it that the arrest of HH signalling in a portion of the cells that repair the rib impacts the entire differentiation program of this population?

Request for further explanation regarding a potential mechanism of the *Sox9*+ lineage cells:

Although there are a number of studies published that claim to identify skeletal stem cells in mammalian systems, our study is the first to show, using genetic approaches, that a specific cell population (within the Sox9+ lineage) has a critical and required role in large-scale bone repair. We further provide evidence that this subpopulation coordinates repair by inducing neighboring cells to differentiate and build a bridging callus. To date the identification of a population that has this role in bone repair has never been described, thus we believe our study merits publication based on these novel findings. Determining and testing the mechanism by which Sox9+ lineage cells signal to other cells is a major project (ID signal, test genetically its requirement) and we determined that it would be out of scope for this initial publication. However, in our initial submission, we proposed the idea that the signal propagated by Sox9+ lineage cells was another round of Hh signaling. Thus, in our revised version we have followed-up to determine if this hypothesis is likely correct by examining when Hh ligands are expressed, if Sox9+ lineage cells release a Hh signal, and if neighboring cells showed evidence of having received and responded to a Hh signal. We used RNA in situ hybridization to detect the expression of *Shh*, *Ihh*, *Gli2* and *Ptch1* combined with Sox9+ lineage analysis. Based on these experiments we conclude that Sox9+ cells are not likely to be emitting a second wave of Hh signals and that they coordinate repair via some other to-be-identified non-autonomous mechanism. The data in support of this conclusion are now included in a new figure, Figure 3. In addition, we have now changed our thinking regarding the function of the Sox9+ lineage cells and this is reflected in completely revised portions of the discussion. After examining the location of Sox9+ lineage cells in relation to regions in the callus undergoing differentiation, Sox9+ cells appear dispersed and not likely to undergo differentiation prior to other neighboring cells in the callus. We have thus changed the title to reflect a role for these cells as critical “messengers” during the repair process rather than "pioneer" cells.

2) Is Hh signaling uniquely involved in regeneration or simply necessary in general for ossification?

Request to determine if Hh signaling is uniquely involved in regeneration or simply necessary in general for ossification:

We believe that the role of Hh in large-scale rib repair is distinct from its role during endochondral ossification. In the growth plate, Hh is required for proliferation of developing chondrocytes and for the maturation of osteoblasts into osteocytes. During rib repair we do not find evidence to support a role for Hh in proliferation. Instead, we provide evidence that Hh is required specifically in the Sox9+ lineage to stimulate their activity as messenger cells. Further, to address this concern, we now include results from a new Tamoxifen regimen in which we ablate *Smo* several days (3-5 dpr) after injury, to evaluate the role of *Smo* “Late” in differentiating callus cells. In agreement with Baht *et al.,* 2014 although some delay in repair is evident, we see robust repair and bone formation at 14 dpr, while in contrast, removal of *Smo* earlier results in no repair at this stage. Thus, we believe that the critical requirement for *Smo* is early in repair. Furthermore, although bone production is somewhat delayed, our results support the idea that *Smo* is not strictly required for generating bone.

3) Problems with our description of the experimental regimens:

In reviewing all the comments, it became clear that we did not adequately explain the design of our experiments and their rationale. We feel that this led to some confusion regarding how we arrived at our conclusions. To remedy this, we have re-written portions of the paper that describe the experimental set-up, particularly with regard to the different timings of tamoxifen administration and what cells are affected as a consequence. We have also used visual cues to orient the reader to the figures and we have completely re-written the legend to Figure 6 (model). We hope that these changes make the paper more accessible to both readers in and outside the field of regeneration biology.

Separate reviews;

Reviewer #1:

In this work, Kuwahara et al., examine rib regeneration using a murine model. They show that the rib regenerates from cells located in the periosteum and that Hedgehog signaling is required for the initial differentiation of the cells that will comprise the regenerating tissue. These cells are derived from a population of cells that express Sox9, and the appear to give rise to cells that co-express Col1 and Col2. They propose a model whereby a rare population of Sox9+ cells in the periosteum recruit other cells to participate in the regenerative process by making the "hybrid cells" that co-express Col1 and Col2. The major finding of this work are two-fold. First, what appear to be chondorcytes co-express Col1 and Col2-markers of osteoblasts and chondrocytes respectively. 2nd that deleting Smoothened from a Sox9+ population of cells in the periosteum leads to poor healing and indicating that hedgehog signaling is required at an early stage of cell differentiation. While this is interesting some fo the conclusions are not supported by the experimental approach and resulting data, and there are some concerns about the exclusive use of in situ hybridization to demonstrate the presence of hybrid cells.I disagree that mammals exhibit a limited capacity for bone repair. In fact, bone is one tissue that exhibits excellent regenerative potential in mammals. The entire basis of distraction osteogenesis is based on the ability of bone to regenerate itself. Further, there is no comparison of healing of the rib to any other bone in the body in this work. So saying the ribs are a "notable exception" is not necessarily correct.

We agree that bone can repair quite well as evidenced by distraction osteogenesis treatments. What we meant to convey is that the skeleton does not naturally repair itself when there are very large injuries. In addition, the reviewer is correct to point out that not all bones have been tested for their ability to heal after large injuries. We believe that the repair seen in the rib is unprecedented in scale and considering this reviewer’s comments we have now changed the introductory text to focus specifically on large-scale repair (see Introduction, paragraph one).

The statement that not all cells in the callus are derived from Sox9-expressing progenitor cells is not supported in the literature (He et al., 2017). In fact, Figure 2E shows that by extending the labeling protocol to 2 days after injury, there is a large contribution of Sox9+ cells to the regenerating tissue.

We now realize that we need to be more explicit about our experimental design. In our study, we show the outcome of two different tamoxifen regimens. When induction occurs prior to injury we observe that only some cells in the callus are derived from a population that expressed *Sox9* prior to injury. This observation is indeed supported by He *et al.,* 2017 where the *Sox9*-expressing cells are also labelled with Tamoxifen injection prior to injury. Indeed we are grateful for Xinjun He for sharing the results of his experiments prior to publication (downstairs from us) and his results led us to use a similar Sox9-Cre strategy. Similar to our Figure 2C, if the panels of Figure 4 of He et al., 2017 are enlarged, it is clear that there are many callus cells that do NOT carry the tdTomato lineage trace indicating that they are derived from other sources that did not express *Sox9* prior to injury. Although most cells express *Sox9* in the callus as seen by RNA in situ hybridization (our Figure 1B) and by immunofluorescence for *Sox9* protein expression (Figure 4, He et al) this just shows that *Sox9* can be upregulated after injury, not that these cells are necessarily a special population. Additionally, it is known from other studies that other, non-*Sox9* expressing lineages contribute to bone repair such as Pax7-expressing cells from muscle (PMID: 25594525).

In our Figure 2E, we perform a very different experiment than in Figure 4 of He et al., where we induce recombination of the tdTomato reporter before and after injury. This will capture both the cells that expressed *Sox9* prior to injury and the cells the express *Sox9* in response to injury. Thus there are many more cells labelled. At later time points it is true that many cells (~85%) come from the lineage of cells that were expressing *Sox9*. However, upon careful examination, there are some cells (~22%) that are not marked by the tdTomato reporter, indicating that they may come from a lineage that never expressed *Sox9* prior to Tamoxifen induction. In response to these comments, we have made significant alterations to our text in subsection “The murine rib bone regenerates through a hybrid skeletal cell type” to better explain these two results and used visual cues in the figures (see Figure 2) to better distinguish between the two different types of experiments.

Hybrid cells:It would be interesting to assess rib healing when the periosteum is removed along with the bone. Do hybrid cells still appear?

In our previous publication (Tripuraneni et al., 2015, PMID 25651082, Figure 2B’), we showed that no healing occurs when both the bone and periosteum are removed (2 months post healing). Instead interstitial connective tissue forms with no cartilage or bone evident. In response to this comment, we also examined healing at an earlier time point (7 dpr) when normally cartilage would be evident and saw no cartilage formation (Figure 2—figure supplement 1). We then examined *Sox9*/Runx2 expression and did not see an abundance of double positive cells. Thus, we believe that in the absence of the periosteum, early osteochondral progenitors are not present and do not differentiate into cartilage or bone tissue.

Why is there no Col1 expression in the cells embedded in the new trabecular bone?

It is well-known that *Col1* production diminishes significantly as osteocytes mature, so we do not expect to see high expression in these cells. However osteoblasts and new osteocytes express *Col1* and this is evident in Figure 1D, Figure 1—figure supplement 2, and Figure 5A. To illustrate this more clearly, we have adjusted the area enlarged for Figure 5A which clearly shows cells embedded in matrix expressing *Col1* (marked with arrowheads).

Do the hybrid cells co-express Col1 and Col10? Or just Col1 and Col2? It is hard to assess this from the figures.

We believe that callus cells express cartilage and bone genes in a similar sequence to developing cartilage and bone cells. We do observe co-expression of *Col10* with *Col1* RNA (data not shown) and we also observe expression of both the chondrocyte *Col10*mcherry reporter and the osteoblast specific reporter (*Col1(2.3) GFP*). This is now shown in Figure 1E.

Hedgehog experiments:Shh expression has been examined in other work on rib fracture models (Matsumoto et al., 2016), and it appears the results are different from those here. This should be discussed.

We also see expression of Shh early in the repair callus of the rib as previously reported. A reference to Matsumoto et al. is now included and our data in support has been added to Figure 3A, C.

Where is Shh, Ihh, Ptc expressed prior to chondrocyte differentiation? It seems this is a key piece of data that is missing. Instead Figure 3 shows expression of these genes in the chondrocytes. However, knock-out of Smoothened affects an earlier stage of healing.

We now include our data examining the expression of Shh, Ihh, *Gli2*, and Ptch1 when cells are in the process of differentiation in Figure 3A, C and Figure 2—figure supplement 1A, C.

Work by Baht has already shown that Hedgehog signaling is dispensable during expansion of the cartilage in the callus. Thus, there is a disconnect between the expression patterns of Shh, Ihh, Smo, and Ptc, and the experimental data.

We do not know why in Baht et al. the cartilage callus still forms. One possibility is that the Smo KO is not actually complete (no confirmation is shown and at least for the *Col2*Cre line, we have been unsuccessful in obtaining recombination from this transgene during repair). Another idea is that there are differing requirements for Hh signaling depending on the type of injury (fracture vs. large-scale repair). We now include these ideas in the Discussion section (fourth paragraph). We also now include all the expression patterns to show the activity of the Hh pathway early in repair (see response to comment 8).

Model:In Figure 5, I do not see evidence in the manuscript of Sox9+ osteocytes or hybrid cartilage cells that do not express Sox9. Also, it is not clear why there are Sox9- hybrid cells since when lineage tracing was extended to after the injury, nearly all of the regenerated tissue was derived from labeled cells (Figure 2E). It seems that many more cells express Sox9 after injury, which could just reflect activation of a larger number of the stem cell pool by the injury itself, and not reflect a recruitment of these cells by the prior Sox9+ population as discussed in paragraph one of the Discussion section. To show that these cells act as pioneers that recruit other cells, the Sox9+ cells would need to be depleted prior to injury and healing examined.

One of the most interesting results from our work is that the KO of Smo in a small population that expresses *Sox9* prior to injury (this population only contributes to a minority of the callus) results in a severe phenotype (no healing). These are the results we wished to highlight and explain in the model diagram. We hypothesize that these *Sox9*+ lineage cells are important coordinators of repair and influence neighboring cells through non-autonomous mechanisms to adopt a hybrid cartilage/bone character. These neighboring cells may have originated from the periosteum but not from the subpopulation that expressed *Sox9* prior to injury and so many of them are not depicted in red in the model diagram (matching our lineage tracing results in Figure 2C). Under the influence of signals from the *Sox9*+ lineage cells, we propose that these neighboring non-*Sox9*+ lineage cells differentiate, adopt a hybrid cartilage/bone character, and build a bridging callus. We realize our original description was perhaps not clear, and we have now added more a detailed explanation in the legend to a revised Figure 6.

Minor Comments:Figures 2 and Supplemental 2 should be combined. It is difficult to compare the location of the images when the histology and in situs are in separate figures.

We have now done as suggested.

Additional data files and statistical comments:An additional method of assessing co-expression of Col1 and Col2 should be considered, either PCR after FACS, immunohistochemistry, or an alternate method. Since this is a central tenet in the paper, a more robust analysis would be useful.

We now include additional evidence with immunofluorescence for COL1 and COL2 in Figure 1—figure supplement 1D. We also now include an experiment using two transgenes that mark hypertrophic chondrocytes (*Col10a1*-mCherry) and osteoblasts (*Col1(2.3)-GFP*) in Figure 1E. Both methods show results consistent with our RNA-ISH results.

Reviewer #2:

[…] I have the following specific comments:1) During endochondral ossification in the growth plate, SOX9 levels are known to vary in distinct cartilage cell types at various stages of differentiation (Leung et al., 2011, PMID 22072985). Therefore, the authors need to justify that all Sox9+ cells of the periosteum are the pioneer population, or they would have to use another marker (and another Cre line, see below).

This reviewer makes the excellent point that it is possible that only a subset of the *Sox9*+ lineage population (already only ~6% of the periosteum) are the critical players. We have acknowledged this possibility in the Discussion (paragraph two). Further experiments will need to be done to determine if there are subgroups within the *Sox9*+ population that have a higher healing potential. Unfortunately, the tools to address this question are not available and we feel it is reasonable at this time to conclude that *Sox9* expression is so far our best indicator of a cell type in the periosteum that is critical for large-scale rib repair.

2) The authors are using a Sox9CreERT2 line to lineage trace the source of cells that give rise to the callus. It is unclear from the data, if most cells are recruited from the region neighboring the injury site or from more distant cells.

We are not sure which experiment this question is referring to (Pre or Post regimen). Considering our Pre experiment, we cannot definitively determine if most *Sox9*+ lineage cells come from a region neighboring the injury site or from a more distant location without doing real-time in vivo imaging. We hypothesize that most arise from the nearby periosteum, since without the periosteum, repair fails (Tripuraneni et al., 2015 and Figure 2—figure supplement 1).

3) Other periosteum/osteoblasts specific CreERT2 lines should have been used to validate the periosteum as the source of cells in the callus.

Previous lineage tracing experiments have shown that the majority of the callus comes the periosteum (example: Colnot, 2009). We also now show that when the periosteum is removed along with the rib bone, there is a failure to build a callus (Figure 2—figure supplement 1). Together these results indicate that the periosteum is a main source of cells in the callus.

4) The authors mention in their opening statement that "most bones in mammals display limited capacity for repair". Is it known if there is also a reduced number of Sox9+ cells in other bones which have limited repair capacity?

With help from Dr. Jay Lieberman’s lab we have now added a comparison to the femur which has more limited repair capacity. We find that the number of *Sox9*+ cells in this bone, either before and after fracture is not significantly different from our findings in the rib. These results are shown in Figure 2—figure supplement 2A. Thus some other property either intrinsic to the cells or related to their environment may explain differences in repair capacity.

5) Does hedgehog signal in an autocrine rather than paracrine manner? It would be good to know if only a subset of cells activate Gli or specific Hh target genes.

Based on this interesting question we have carried out further analysis in both controls and Smo KO animals looking at target gene expression. Interestingly, we still see strong Shh expression early in Smo KO animals in many cells across the callus (Figure 3C). Neither Shh nor Ihh expression could be found specifically or particularly strongly expressed in Tdtomato+ cells (there is some minor overlap). Thus the characteristics of the population that releases a Hh signal is yet to be determined. In addition, we believe that the expression of Ihh and Shh later in chondrocytes is more reflective of their advanced differentiation and perhaps a role in promoting osteocyte maturation rather than in stimulating *Sox9*+ cells to mediate repair. Indeed using an additional new experimental tamoxifen regimen now added to the paper, “Late” Smo KO (strictly after injury) we did not observe a dramatic effect on repair (some delay evident but complete bridging still occurs), Figure 5—figure supplement 1. Additionally, in control animals, we did not find that cells nearby Tdtomato+ cells had higher Ptch1 or *Gli2* expression, Figure 3C and Figure 3—figure supplement 1A, C. All together, these results suggest that in contrast to our initial hypothesis, the predominant orchestrating signal released by the *Sox9*+ cells is not a HH ligand. Some of these ideas are now included in the discussion.

6) This reviewer was not convinced by the studies implicating Hh signaling in regeneration. Although the authors observe that disruption of Smo leads to a reduction in Col2a1/Col1a1 expression in the callus, hedgehog signaling cannot have a unique role in regeneration since it also has a very general role in endochondral ossification.

Although Hh signaling is known to have a role in endochondral ossification (specifically controlling cartilage cell proliferation, Long et al., 2001) we argue that its most important role in regeneration is distinct in inducing the *Sox9*+ lineage cells to become messenger cells that coordinate repair by inducing neighboring cells to differentiate into cartilage/bone hybrid cells that mediate the repair process. We have replaced the word “unique” and with the word “distinct” to be more accurate.

Reviewer #3:

In this paper, Kuwahara et al. investigate the regeneration of rib bones in mice, using a model of rib ablation. They characterized the population of cells, all derived from the periosteum, which repair the resected rib. Using ISH they found, within the repairing callus, cells (which they name "hybrid" skeletal cells) that co-express chondrocyte-specific and osteoblast-specific markers. Using lineage tracing techniques and Sox9 as a marker of the chondrocyte lineage, they find that the callus comprises a minor (1/3) population derived from cells expressing Sox9 in the periosteum, while the majority (2/3) upregulate it after resection. The ablation of Smo in the minor population or in the entire population leads to a similar phenotype, defects in callus formation and bone differentiation. The authors conclude that the periosteum-derived Sox9 cells play an essential function in the HH-mediated bone regeneration of ribs.In general, the study is interesting and it could be worth publishing in a good journal. The illustrations are of excellent quality, even though a little more infos and enlargements would sometimes be helpful. Below are some comments about the text and the terminology used. In addition, there are a few technical points that could rather easily be addressed. The main concern with the manuscript is that a mechanistic understanding of the phenotype obtained after deletion of Smo in the two experimental paradigms is lacking. How is it that the arrest of HH signalling in a portion of the cells that repair the rib impacts the entire differentiation program of this population? Without this, the study seems a little preliminary.Text and terminology:Why the question that is addressed here is worth studying and what are the key findings of the study could be explained more clearly in the Introduction. For non-specialists it takes a lot of reading of this and many other papers (and there are many) to understand what is novel about this study.

Thanks for this comment, we completely re-worked the Introduction with the aim of making it more readable to a broader audience.

The terms "pioneer" and "hybrid" are odd.Pioneer: that is first to explore or to do something novel: Darwin and Mendel, certainly; pioneer neurons, pioneer muscle cells, maybe. But here it does not seem justified.Hybrid: characterizes the progeny of two distinct breeds, species. Wouldn't it be more correct to call them bi-potent progenitors?

We initially called these cells “pioneer “cells because we thought they might migrate in first, and they appeared to lead the repair process by both differentiating and also emitting a differentiation signal. However, based on experiments suggested by the reviewers, we now believe, that while they may migrate in early, they don’t appear to be the first to differentiate (Figure 3—figure supplement 1B). We have therefore changed the term to “messenger” to illustrate their important coordinating and orchestrating role.

We used the term “hybrid” because the cells express typical cartilage- and bone-specific genes simultaneously. The term “bi-potent” implies that they will differentiate into either bone OR cartilage and in this repair context, we believe the fate of the cells is predominantly osteocytes.

We could not think of a suitable substitute word. Thus, to make sure there is no confusion, we have carefully defined the term and continue to qualify it using the word “osteochondral” throughout the manuscript.

"Using the tdTomato reporter, we confirmed that the Pre tamoxifen treatment resulted in deletion of Smo in tdTomato+ cells within the resection site. In contrast, the Post tamoxifen treatment resulted in deletion of Smo throughout the resection site (Figure 3—figure supplement 1C)"These phrases are unclear: the authors oppose "within" and "throughout". What is the difference? Explain better.

We have now made large changes to the text which should help clarify the different outcomes (subsection “Lineage tracing of Sox9-expressing cells to the regenerating rib callus”).

Pictures:Enlarged pictures (maybe insets) of the Figures 1B and C with indications to identify the mesenchymal-like and cartilage-like cells would help non-specialists.

We have now added insets as recommended.

Technical improvements:Some ISH data are a little confusing. First, they are done at different times: 5 dpr in Figure 1 and 7 dpr in Figure 1—figure supplement 1. Adding to the confusion is the fact that the lineage analyses are started at 10 dpr, such that it is difficult to compare ISH and lineage analyses.

Our goal was to show that cells with dual cartilage/bone properties persist throughout the repair process. We could show earlier lineage analysis but couldn’t determine how this would add to the study. Instead we hope that some new phrasing of the text (2nd paragraph of the results) helps with understanding the data.

Second, the Sox9 fluorescent ISH indicates that just about all cells of the callus are positive (Figure 1 —figure supplement 1A), while a small proportion of them are positive using a colorimetric approach (Figure 1—figure supplement 2). Could you comment this discrepancy?

In general, we and others have found that the colorimetric assay is less sensitive than the fluorescent one. Additionally, although the sequence of events during repair is highly reproducible, we do see some variation in timing. Thus, although both samples are shown at 7 dpr, the repair callus in Figure 1—figure supplement 2 may be slightly more mature than the callus in Figure 1—figure supplement 1A.

"To understand whether cells also upregulate Sox9 expression during callus formation and to also potentially capture more cells within the callus, we instead applied tamoxifen at the time 18 of injury plus 2 days following".This seems to be an obscure reason to do this experiment. It would be more appropriate to justify the Post- regimen experiment by doing and ISH at 10 dpr and determine whether there is a discrepancy between the number of Sox9-derived cells after the Pre- regimen and the number of Sox9+ cells as determined by ISH. According to the ISH shown in Figure 1 and Figure 1—figure supplement 2, there should be a major difference between the two which would justify that the authors utilise another approach.

We agree that the experiment suggested by the reviewer is better way to determine if *Sox9* is upregulated after injury and thus the justification outlined in the text for the “Post” experiment doesn’t work. We have now changed the lead-in to this section of the paper.

Mechanism:Since it is clear from the presented data that HH signalling is implicated in callus formation and in the downstream differentiation, the authors need to find an explanation for the similarity between the phenotype obtained after Pre- or Post- Smo ablation. At this stage, such mechanism is lacking.

We believe that there is no difference in phenotypic outcome between the Pre and Post Smo ablation because the Sox9+ lineage population is the key cell type that requires Hh signaling. Both Tamoxifen regimens would result this subpopulation being null for Smo KO. Based on both these outcomes we hypothesize that in response to Hh signaling, Sox9+ lineage cells produce a second differentiation signal that induces neighboring cells to differentiate into the cartilage repair callus. We have now improved the text describing these ideas in the discussion and in the legend to model Figure 6.

Are the Smo+ cells still present after the Pre- protocol the same ones (2/3) than the ones that activated Sox9 after resection? If yes, this would indicate that HH signalling is not implicated in the recruitment of Sox9 positive cells by the periosteum-derived population.

Yes, the cells expressing *Smo* likely also express *Sox9* after resection since both are broadly expressed throughout. We did not mean to imply that Hh signaling is involved in the physical recruitment or migration of the cells. Our hypothesis is that Sox9+ lineage cells require Hh signaling in order to produce another signal (to be identified) that induces the differentiation of neighboring cells (including those expressing *Smo* and *Sox9* after injury). We have made this idea clearer in the Discussion text (last paragraph) and the legend of Figure 6.

The presence of Smo does not mean that the pathway is active. To check this, the authors should look at Ptc (as a sign of a response; maybe Gli could be used as well) and HH expression (to determine whether the signal is there or not). If Ptc and HH are normal, then they should find another reason explaining this.

Thank you for these suggestions. We have added expression data on *Ptch1* and *Gli2*, as read-outs of Hh signaling, Figure 3C and Figure 3—figure supplement 1A, C We still see strong *Shh* expression early in Smo KO animals in many cells across the callus (Figure 3C). Neither *Shh* nor *Ihh* expression could be found specifically or particularly strongly expressed in Tdtomato+ cells. Additionally, in control animals, we did not find that cells nearby Tdtomato+ cells had higher *Ptch1* or *Gli2* expression, Figure 3C and Figure 3—figure supplement 1C. In summary these results suggest that the predominant signal released by the Sox9+ cells is not a HH ligand. These ideas are now included in the discussion.

[Editors' note: further revisions were requested prior to acceptance, as described below.]

The manuscript has been improved but there are some remaining issues that need to be addressed before acceptance. Specifically, the analysis of the Hh pathway is incomplete and this is a weakness of the paper. The authors have still not addressed the fact that the majority of the cells that co-express Col2a1 and Col1a1 are not Sox9 lineage. This is really important for the entire premise of the paper. In addition, there are other smaller points raised by the reviewers that you can see in the full reviews included below.

Reviewer #1:

This work still lacks an appropriate analysis of the Hh pathway at the time that the Sox9+ cells are initially stimulated in the periosteum. Since this is the model that is being put forth, it is necessary to show how/where this pathway is working at this early time point. While this was pointed out previously, the authors have done a very thorough job addressing the other concerns.

We now include analysis at 3 dpr (Figure 3—figure supplement 1A) showing that *Sox9*+ cells in the periosteum express the Hh pathway readout, Ptc1. We believe that *Sox9*+ cells respond to Hh signaling while they are still in this peripheral location. As they migrate into the callus their necessity to respond to Hh appears to decline.

Minor comments:It is confusing why this sentence: "Instead, Sox9-positive lineage cells require Hh signaling to stimulate neighboring cells to differentiation [sic-typo] into a skeletal cell type with dual properties of chondrocytes and osteoblasts" remains in the Abstract, since in the response to reviewers the authors state: "we conclude that Sox9+ cells are not likely to be emitting a second wave of Hh signaling.…" This seems at odds and could be made more clear.

We have now changed the Abstract to make it more clear that we think the second wave of signaling is not yet known.

In subsection “Lineage tracing of Sox9-expressing cells to the regenerating rib callus”, the statement that "cells within the femur fracture callus expressed Col1a1 and Col2a1" needs to be modified to "some cells," because from the image it appears the vast majority of cells express one or the other of these genes, but not both. There are no negative controls shown to determine levels of background fluorescence.

We have now changed the text to be more conservative to say “some” as recommended.

Reviewer #2:

The authors have improved the manuscript, however I still have a major concern that remains unaddressed. If we compare 14 dpr PRE-induction schedule results in Figure 5A CONTROL and Figure 2C (which determine the contribution of Sox9 lineage at an early 14 dpr stage), then it seems that the majority of the cells that co-express Col2a1 and Col1a1 are not Sox9 lineage. This same concern I had pointed out earlier when I earlier said "cells are recruited from the region neighboring the injury site or from more distant cells". This is not been clarified in the discussion in the current version of the manuscript.

This is indeed correct and a point that is very important in the paper. We believe that Sox9 lineage cells emit a yet-to-be identified signal that induces neighboring cells to participate in building the callus and to start expressing differentiation genes of cartilage and bone. We have now included additional text throughout the manuscript to drive-home this point.

This has further consequences if we now address the role of Hh signaling (compare Figure 5 PRE- and POST-induction schedule) in large-scale repair. In both PRE- and POST-induction schedule Smo mutants, none of the cells co-express Col2a1 and Col1a1; this suggests that the major contribution is from an altogether different source/ or an extended periosteum which was laid down post trauma/ etc. This point also needs attention.

Yes, the majority of the cells that build the callus must come from somewhere else. We believe they mainly come from the non-*Sox9* lineage populations in the periosteum but could ostensibly come from a variety of sources. We have now included this idea in the Discussion.

To address this issue, the authors might want to revisit the Sox9CreERT2 lineage tracing. The percentage contribution can be determined for 10, 14 and 21 dpr for both PRE- and POST- induction schedule.

After the bony callus is made there is a remodeling phase where bone is degraded and replaced. At 14 dpr, remodeling has begun and many of the tdTomato cells are beginning to be remodeled out. This was also observed post-fracture by He et al., 2017 (comparing 9 days to 2 months post-femur fracture in Figures 4 and 5). We now include this idea in the manuscript.

In Figure 2C, there seems to be higher tdTomato+ cells in the regenerated region at 10 dpr that 14 dpr (magnification in 21 does not match the others.)

To avoid confusion, we have now moved the images of the older samples to supplemental as the point they were intended to make— that there is longer-term contribution of *Sox9*-lineage cells with osteocyte-like morphology (hence the higher magnification) to the repair site— is tangential to the paper, and has also already been shown post-femur fracture by He et al., 2017, Figure 5.